# KO: Kinetics-inspired Neural Optimizer with PDE Simulation Approaches

## Abstract

The design of optimization algorithms for neural networks remains a critical challenge, with most existing methods relying on merely heuristic adaptations of traditional gradient-based approaches. This paper introduces KO (Kinetics-inspired Optimizer), a novel neural optimizer gadget inspired by kinetic theory and partial differential equation (PDE) simulations. KO can be used with multiple types of base optimizers (e.g., Adam, SGD). In KO, the training dynamics of network parameters are perceived as the evolution of a particle system, where parameter updates are simulated via a numerical scheme for the Boltzmann transport equation (BTE) that models stochastic particle collisions. This physics-driven approach inherently promotes parameter diversity during optimization, mitigating the phenomenon of parameter condensation, i.e. collapse of network parameters into low-dimensional subspaces. Parameter condensation is proven harmful to model generalizability. We analyze KO's impact on parameter diversity, establishing both a strict mathematical proof and a physical interpretation. The convergence of the proposed optimizer can also be guaranteed. Extensive experiments on image classification (CIFAR-10/100, ImageNet) and text classification (IMDB, Snips) tasks demonstrate that KO consistently outperforms baseline optimizers, achieving accuracy improvements while remaining comparable computation cost.

## 1 Introduction

Optimization algorithms are the cornerstone of training modern neural networks, directly influencing model convergence, generalization, and robustness. Gradient-based methods, such as stochastic gradient descent (SGD) (Robbins & Monro, 1951) and adaptive learning rate optimizers like Adam (Kingma & Ba, 2014) and AdamW (Loshchilov & Hutter, 2017), dominate deep learning practice due to their simplicity and empirical efficacy (Devlin et al., 2019; Bai et al., 2023; Liu et al., 2024a). Concurrently, interdisciplinary efforts have explored physics-inspired optimization paradigms, such as thermodynamic (Moein & Logeswaran, 2014) and electromagnetic field analogies (Abedinpour-shotorban et al., 2016), which interpret network training as energy minimization of dynamical systems. These approaches borrow conceptual tools from classical mechanics or statistical physics to boost methods' interpretability and performance.

Despite their widespread adoption, existing optimizers suffer from a critical limitation: they lack mechanisms to systematically regulate the distribution of network parameters during training. Empirical studies (Zhou et al., 2022a; Chen et al., 2023) reveal that parameters in over-optimized models often condense into low-dimensional subspaces, a phenomenon termed parameter condensation (Xu et al., 2025). Parameter condensation often leads to diminished model generalizability (Jin et al., 2020). While regularization techniques like weight decay or dropout partially mitigate this issue (Zhang & Xu, 2024), they act as post hoc corrections rather than addressing the root cause in the optimization dynamics. Furthermore, current physics-inspired methods often oversimplify the analogy between physical systems and neural networks, neglecting the stochastic, many-body interactions that govern parameter evolution. This gap raises a pivotal question: Can we design optimizers that inherently promote parameter diversity by simulating the microscopic, collision-driven dynamics observed in kinetic systems?

This work introduces KO, a kinetics-inspired neural optimizer gadget that reformulates parameter training as the evolution of a particle system simulated via partial differential equations. Drawing parallels between network parameters and particles undergoing collisions, KO employs a numerical

discretization of the BTE (Boltzmann, 2015) to model stochastic parameter interactions. This approach naturally enforces parameter dispersion through a thermal diffusion-like mechanism, directly counteracting condensation tendencies. The highlights of this work are threefold:

1. **Physics-Driven optimization framework:** We propose the first optimizer, to our best knowledge, that rigorously integrates kinetic theory and PDE simulation into neural network training, bridging microscopic particle dynamics with macroscopic learning outcomes.

2. **Theoretical and empirical analysis:** We prove that KO's collision-based updates decrease the weight correlation, and we interpret this behavior by both mathematical and physical analysis. KO's convergence is also guaranteed through strict proof.

3. **Empirical performance superiority:** Experiments across image and text classification tasks demonstrate that KO achieves consistent accuracy gains over classical optimizers.

## 2 RELATED WORK

**Deep-learning Gradient-based Optimizers.** The advancements in optimization algorithms has been the key to boosting deep learning. Early methods such as Stochastic Gradient Descent (SGD) established the foundation by updating parameters along gradient directions (Ruder, 2016). On top of that, Nesterov (1983) introduces the momentum mechanism, enhancing convergence by incorporating past gradients. The development of adaptive learning rate methods marked a major shift, with approaches like AdaGrad (Duchi et al., 2011), RMSProp (Tieleman & Hinton, 2012), and Adam (Kingma & Ba, 2014) adjusting step sizes based on gradient history. Second-order methods like BFGS (Fletcher, 2000) and L-BFGS (Liu & Nocedal, 1989) further improve convergence by approximating the Hessian matrix at a larger computational cost. Recent advancements include LAMB (You et al., 2019), which scales learning rates based on layer-wise weight norms, and AdaFactor (Shazeer & Stern, 2018), which reduces memory usage in large models. These developments underscore the ongoing evolution of optimization techniques in deep learning, driven by the need for efficient and effective training methods.

**Physics-Inspired Metaheuristics Optimizers.** Researchers have taken inspiration from non-linear physical phenomena to formulate meta-heuristic optimization algorithms. For instance, gravitational search algorithms (Formato, 2007; Rashedi et al., 2009) design searcher agents as a collection of masses that interact with each other based on Newtonian gravity and the laws of motion. Likewise, the electromagnetism-based algorithms (Javidy et al., 2015; Abedinpourshotorban et al., 2016; Yadav et al., 2019) define agents as charged particles or electromagnets driven by the electrostatic or magnetic forces. Other metaheuristics include the fluid Mechanics (Doğan & Ölmez, 2015; Tahani & Babayan, 2019), optics (Kaveh & Khayatazad, 2012; Kashan, 2015), thermodynamics (Moein & Logeswaran, 2014; Kaveh & Dadras, 2017) and nuclear physics (Wei et al., 2019). These algorithms have been successfully applied to various non-linear optimization problems. However, they are not directly applicable to general deep learning optimization due to the gradient-free nature and the lack of a clear connection between the physics and the gradient-based network optimization dynamics.

Another line of research explores physics-inspired optimization through mean-field-based frameworks. Mei et al. (2018) analyze the dynamics of SGD using distributional dynamics governed by partial differential equations, while Rotskoff & Vanden-Eijnden (2022) take a sparse perspective, modeling particle interactions via potential functions. Our work builds on this sparse particle system viewpoint, but distinguishes itself by adopting collisions as the primary mode of interaction.

## 3 METHODOLOGY

### 3.1 PRELIMINARIES: KINETIC THEORY AND NUMERICAL ALGORITHM

The kinetic molecular theory of ideal gases (Loeb, 2004) is built on several simplifying assumptions. First, a gas is composed of identical particles called molecules. Second, these molecules are in constant motion, and follow Newton's laws of motion. Third, the molecules act as small, elastic spheres with negligible volume. This implies that the collisions within the system are perfectly energy-conserving. Finally, interactions are limited to collisions, with no long-range attractive or repulsive forces.

While these assumptions provide a useful microscopic picture, directly tracking the motion of individual particles becomes intractable in systems with an extremely large number of molecules.

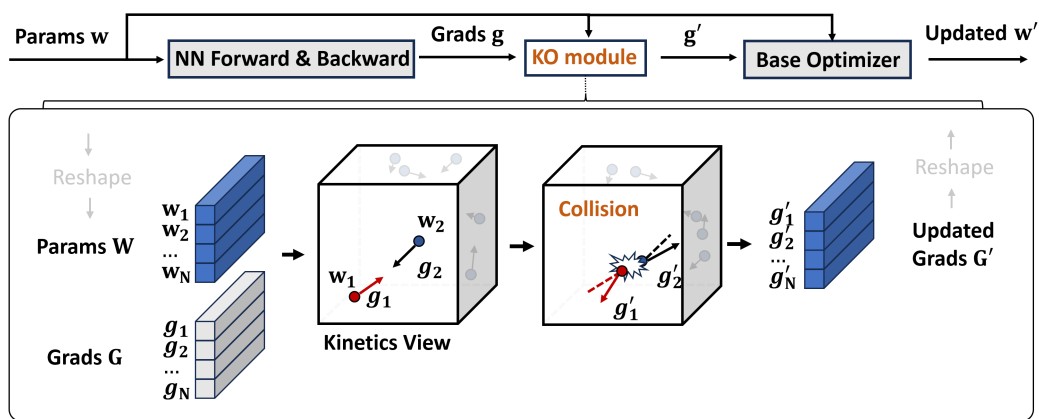

Figure 1: The architecture of KO. The upper part is the workflow of KO composed of a kinetic module and a base optimizer. The gradient is first calculated by network backpropagation, and then updated by the kinetic module, and finally fed into the base optimizer. The lower part is the kinetic module, which simulates the particle collision to update the gradient. The weight and gradient are viewed as the position and velocity of the particles with random collisions. Once two particles collide, the velocity of the particles is updated by the hard-body collision model.

To address this, one instead turns to the statistical distributions to describe the system's dynamics. The change in the number of particles $N$ is hence characterized by $f$, a density function defined in a 7-dimensional phase space: $dN = f(\mathbf{x}, \mathbf{p}, t)\, d^3\mathbf{x}\, d^3\mathbf{p}$. If position $\mathbf{x}$ and momentum $\mathbf{p}$ evolve according to the Hamiltonian equations under an external force $F_{ex}$ on the system, then the density function $f$ satisfies the Boltzmann Transport Equation (BTE): $\frac{\partial f}{\partial t} + \frac{\mathbf{p}}{m} \cdot \nabla_{\mathbf{x}} f + F_{ex} \cdot \nabla_{\mathbf{p}} f = \left( \frac{\partial f}{\partial t} \right)_{coll}$, where the right-hand term captures the distribution changes caused by particle collisions. Since no closed-form expression exists for this term, it is typically approximated empirically. As a partial differential equation, BTE governs the temporal evolution of the distribution function $f$. The Direct Simulation Monte Carlo (DSMC) method (Ivanov & Rogasinsky, 1988) and the lattice Boltzmann method (Krüger et al., 2017) are two common numerical approaches for solving BTE.

## 3.2 DIRECT SIMULATION MONTE CARLO (DSMC)

The DSMC is a stochastic approach that solves BTE for dilute gases by simulating particle motion. In DSMC, each simulation particle represents $F_N$ physical molecules. The method discretizes the computational domain into cells, then tracks particle evolution through drift, wall collision, and particle collision in each cell.

During the drift phase, particles follow straight-line trajectories determined by their current velocities, with the assumption that no collisions occur during this phase. When particles hit domain boundaries, the wall collision step processes these interactions according to the boundary conditions. These two steps provide deterministic updates to the system.

Stochasticity is introduced in the particle collision step. Particle collision only happens between particles that fall into the same cell. The hard sphere model, where the collision probability is proportional to the relative velocity of the particle pairs, is utilized to model the collision:

$$P_{\text{coll}}[i, j] = \frac{|\mathbf{v}_i - \mathbf{v}_j|}{\sum_{m=1}^{N_c} \sum_{n=1}^{m-1} |\mathbf{v}_m - \mathbf{v}_n|}, \quad (1)$$

where the $N_c$ is the number of particles in the cell and $\mathbf{v}$ is the particle velocity. The DSMC method utilizes a rejection sampling approach to efficiently approximate collision probabilities, as direct implementation of Eq. 1 would be computationally prohibitive. The algorithm begins by estimating potential collision pairs $M_{\text{cand}}$ through the no-time-counter technique: $M_{\text{cand}} = \frac{N_c(N_c-1)F_N \pi d^2 v_r^{\max} \tau}{2V_c}$, where $d$ is the particle diameter, $v_r^{\max}$ is the maximum relative velocity, $\tau$ is the time step and $V_c$ is the cell volume. Given this, we can randomly select $M_{\text{cand}}$ pairs of candidate

particles and determine whether the collision will occur given the following criterion:

$$|\mathbf{v}_i - \mathbf{v}_j|/v_{\mathrm{r}}^{\max} > \Re_1. \tag{2}$$

where $\Re_1$ is sampled from the uniform distribution $U(0, 1)$. For accepted collisions, particle velocities are modified according to the chosen collision model without updating the spatial positions of the particles. This cell-wise procedure iterates until all potential collisions are processed.

The simulation employs the hard sphere collision model, which approximates particles as perfectly rigid bodies. This model strictly conserves both momentum and kinetic energy during collisions while randomizing the scattering direction. Post-collision relative velocities are expressed in the polar coordinate form:

$$\mathbf{v}_{\mathrm{r}}^* = v_{\mathrm{r}}[(\sin\theta\cos\phi)\hat{\mathbf{x}} + (\sin\theta\sin\phi)\hat{\mathbf{y}} + \cos\theta\,\hat{\mathbf{z}}], \tag{3}$$

where $\phi = 2\pi\Re_2$, $\theta = \cos^{-1}(2\Re_3 - 1)$ and $\Re_2$ and $\Re_3$ are numbers randomly sampled from the uniform distribution $U(0, 1)$. If we further denote the center of mass velocity as $\mathbf{v}_{\mathrm{cm}} = (\mathbf{v}_i + \mathbf{v}_j)/2$, then the post-collision velocity can be calculated as:

$$\mathbf{v}_i^* = \mathbf{v}_{\mathrm{cm}} + \mathbf{v}_{\mathrm{r}}^*/2, \mathbf{v}_j^* = \mathbf{v}_{\mathrm{cm}} - \mathbf{v}_{\mathrm{r}}^*/2, \tag{4}$$

### 3.3 KINETICS-INSPIRED NEURAL OPTIMIZER

The intuition of the kinetic-inspired neural optimizer is to hinder a phenomenon called neuron condensation of neural network parameters (Zhou et al., 2022a), where neurons in the same layer cluster together during training. The cosine similarity within the network layer is a key metric in evaluating the degree of condensation. We prove in Section 3.5 that excessive condensation obstructs model performance.

The proposed optimizer architecture is visualized in Fig. 1. Our kinetics-inspired neural optimizer is built on a base optimizer, such as SGD or Adam. We draw an analogy between gradient-based parameter updates in optimization and velocity-driven position updates in particle systems. As discussed earlier, optimization often leads to the formation of parameter clusters. To counter this effect, we introduce collisions at the gradient level to promote separation among neurons. Then the optimizer updates the parameters according to the base optimizers' rules. It is noteworthy that we operate at the gradient level rather than directly modifying parameter updates, which allows us to retain the core behavior of the base optimizer. This design makes our method a modular enhancement: it can be integrated into diverse optimizers without undermining their intrinsic properties.

At a high level, the collisions can be seen as introducing physics-inspired perturbations to the gradients. However, unlike Gaussian noise, which is directionless and purely stochastic, our perturbations are theoretically motivated by the goal of alleviating parameter condensation and enhancing generalization. We can even provide a mathematical proof on the effects of this directional mechanism in the next section. Thus, while both approaches involve gradient perturbation, the underlying principles and effects of our method are fundamentally different. In the following, we will bring about two types of collision formulation.

**Hard Collision.** The hard-body collision is an ideal operation to separate neurons apart. Hard collision updates the gradients by simulating the hard-body collision using the DSMC method. Specifically, the update scheme is detailed as follows:

First, we calculate the relative distance and velocity between neurons, as well as the position and velocity of their center of mass, in order to transform the system into a center-of-mass system:

$$(\mathbf{w}_r)_{i,j} = |\mathbf{w}_i - \mathbf{w}_j|, (\mathbf{g}_r)_{i,j} = |\mathbf{g}_i - \mathbf{g}_j|, (\mathbf{w}_{\mathrm{cm}})_{i,j} = \frac{1}{2}(\mathbf{w}_i + \mathbf{w}_j), (\mathbf{g}_{\mathrm{cm}})_{i,j} = \frac{1}{2}(\mathbf{g}_i + \mathbf{g}_j). \tag{5}$$

Secondly, we derive the formula for the changes of gradients to be applied after the collision, i.e. $\Delta\mathbf{g}$:

$$(\Delta\mathbf{g})_{i,j} = (\mathbf{g}_{\mathrm{cm}})_{i,j} + \frac{1}{2}(g_r)_{i,j}\mathbf{n}_{i,j} - \mathbf{g}_i, \quad (\Delta\mathbf{g})_{j,i} = (\mathbf{g}_{\mathrm{cm}})_{j,i} + \frac{1}{2}(g_r)_{j,i}\mathbf{n}_{j,i} - \mathbf{g}_j, \tag{6}$$

where $\mathbf{n}_{i,j}$ is a unit vector sampled uniformly from the sphere and is uncorrelated with the weight matrix $\mathbf{w}$. We further require $\mathbf{n}_{j,i} = -\mathbf{n}_{i,j}$ to ensure zero-mean collision. This formulation generalizes Eq. 4, where $(w_r)_{i,j}\mathbf{n}_{i,j}$ and $(w_r)_{j,i}\mathbf{n}_{j,i}$ originate from Eq. 3 and represent the relative receding velocity after collision in the center-of-mass frame.

---

**Algorithm 1** Hard Collision Based Gradient Updates

---

1: **Input:** layer weight $\mathbf{w} \in \mathbb{R}^{N \times D}$, layer gradient $\mathbf{g} \in \mathbb{R}^{N \times D}$, hyper-parameters: coll_coef.
2: **Output:** updated layer gradient $\mathbf{g} \in \mathbb{R}^{D}$.
3: Calculate relative properties $\mathbf{w}_r, \mathbf{g}_r$ and center-of-mass properties $\mathbf{w}_{\mathrm{cm}}, \mathbf{g}_{\mathrm{cm}}$ by Eq. 5;
4: Calculate the full velocity change $\Delta \mathbf{g}$ by Eq. 6;
5: Select collision pairs by Eq. 7;
6: Apply velocity and position change by Eq. 8, get new gradient $\mathbf{g}$;
7: **Return** g;

---

**Algorithm 2** Soft Collision Based Gradient Updates

---

1: **Input:** layer weight $\mathbf{w} \in \mathbb{R}^{N \times D}$, layer gradient $\mathbf{g} \in \mathbb{R}^{N \times D}$, hyper-parameters: coll_coef.
2: **Output:** updated layer gradient $\mathbf{g} \in \mathbb{R}^{N \times D}$.
3: Calculate the repulsion force: $\Delta \mathbf{g} = -\cos(\mathbf{w}, \mathbf{w})\cos(\mathbf{g}, \mathbf{g})\mathbf{g}$;
4: Update the gradients: $\mathbf{g} = \mathbf{g} + \mathrm{coll\_coef}\Delta \mathbf{g}$;
5: **Return** g;

---

Thirdly, a collision hyperparameter $\mathrm{coll}_{\mathrm{coef}}$ is introduced to control the collision percentage. Unlike traditional DSMC, where collision only occurs between particles in the same cell, our method allows collisions between any pair of particles. For each pair of neurons $i, j$, the collision occurs if:

$$\frac{(g_r)_{i,j} \cdot (U_r)_{i,j}}{g_r^{\max}} > 1 - \mathrm{coll\_coef}, \tag{7}$$

where $(U_r)_{i,j} = e^{-(w_r)_{i,j}}$, $g_r^{\max} = \max(g_r)$. This equation is adapted from the original collision filter in Eq. 2. $(U_r)_{i,j}$ is introduced as a soft locality constraint to reduce the rate of semantically meaningless collisions. To better control the collision percentage, we further introduce $U_r$ to reflect the effective mean free distance. As the relative distance $(w_r)_{i,j}$ increases, $(U_r)_{i,j}$ decreases, reducing the probability of the collision between neuron $i$ and $j$.

Finally, we update the gradients of the neurons by the aforementioned collision terms. The update algorithm is summarized in Algorithm 1.

$$\mathbf{g}_i^* = \mathbf{g}_i + \sum_{j \text{ in accepted pair } i,j} (\Delta \mathbf{g})_{i,j}, \tag{8}$$

**Soft Collision.** The aim of the kinetics-inspired neural optimizer is to reduce the network layers' weight similarity. In addition to directly simulating the collision during the backpropagation, we could artificially design a repulsion force to achieve similar results. Our intuition is to use the gradients and weights at hand to separate neurons. Unlike prior work that primarily considers similarity based on weight vectors alone, we argue that gradient information is equally crucial. If two neurons have similar weights but divergent gradients, they are likely to diverge quickly during training.

Intuitively, the higher the similarity, the larger the repulsion force should be. Furthermore, the more positively correlated the neurons are, the more negative the repulsion force should be to push the neurons away. A coefficient matrix that is negatively correlated with the neuron cosine similarity is introduced. We further extend the similarity concept to the gradients. If the gradients aren't correlated, then similarity won't rise. Soft collision is only applied to those neurons with both high weight and gradient similarity.

Secondly, we should consider how to formulate the repulsion force. In the gradient descent procedure, the weight moves towards the negative direction of the gradient. If we want a neuron $w_i$ to move in the opposite direction of another neuron $w_j$, we can just make $w_i$ move in the gradient direction of $w_j$. The update scheme of soft collision is presented in Algorithm 2.

### 3.4 FURTHER UNDERSTANDING OF KO FROM A PHYSICAL PERSPECTIVE

To further motivate the introduction of collisions, we provide a qualitative explanation from a physical perspective that our collision mechanism can reduce the weight correlation. In classical statistical

mechanics, the H-theorem (Boltzmann, 1970) characterizes the role of collisions in driving the entropy of a system (Rényi, 1961), where entropy is a measure on the disorder and randomness in a system. The H-theorem is a natural consequence of the Boltzmann Transport Equation, which we draw upon previously. It states that under the molecular chaotic collisions, the entropy increases to a maximum with time.

Drawing an analogy to neural networks, we regard neurons as particles, weight correlation as the condensation level of these particles and gradients as the velocities induced by the tendency towards condensation. High condensation corresponds to low entropy: if all neurons collapse to the same value, the system is fully ordered and entropy reaches its minimum. Applying the H-theorem in this context suggests that weight correlation decreases with training iterations under the effects of particle collisions, i.e. the neuron collisions, just as molecular collisions increase entropy in physical systems. This analogy provides an intuitive justification for employing collisions when aiming to decorrelate weights. Although physical theorems might not be directly transferable to optimization theory, they offer compelling motivation. A rigorous theoretical treatment of collisions in the context of network optimization will be provided in the following section.

### 3.5 FURTHER UNDERSTANDING OF KO FROM A MATHEMATICAL PERSPECTIVE

In this section, we attempt to provide a mathematical proof of the capacity of our methods. Specifically, our method is designed to decrease the weight correlation. Weight correlation is defined as the abstract sum of the ' cosine similarity matrix of the network weights. In fact, Jin et al. (2020) states that the weight correlation influences the models' generalization bounds. The analysis is based on the PAC-Bayesian framework (McAllester, 1999), in which we can bound the generalization error w.r.t. the Kullback-Leibler (KL) divergence between the posterior and prior distributions of the network parameters (Dziugaite & Roy, 2017). Jin et al. (2020) proves the relationship between the generalization error bound and the weight correlation in the following theorem.

**Theorem 3.1** (Jin et al. (2020)). *For a nontrivial network, the decrease in network weight correlation results in the reduction of the KL divergence between the posterior and prior distributions of the network parameters, and hence leads to a tighter upper bound of the generalization error.*

We only need to show that our introduction of the collisions reduces weight correlation to prove our methods' effects. Under mild assumptions, we can have:

**Theorem 3.2.** *Given a small enough learning rate, both soft collision and hard collision reduce the weight correlation of the applied network layer.*

The theorem is natural, as our methods artificially provide a force to separate the neurons. The proof is detailed in Appendix A. If we combine the conclusion with Thm. 3.1, we can state that the collision mechanism lowers the models' generalization error. Hence, by directly reducing weight correlation through collisions, we provide a principled and theoretically grounded mechanism to improve generalization, without altering the core dynamics of existing optimizers.

### 3.6 PROOF OF CONVERGENCE

Convergence is a fundamental criterion in evaluating the reliability of optimizers, as it directly impacts model performance. To ensure that our proposed collision mechanism does not compromise this property, we investigate into a formal convergence guarantee in this section. The detailed proof is provided in Appendix A.

**Theorem 3.3.** *Let $\mathcal{L}$ be a L-smooth loss function and be lower bounded by $\mathcal{L}^*$. Consider the parameter update iterations produced by $\boldsymbol{\theta}_{t+1} = \boldsymbol{\theta}_t - \eta(\boldsymbol{g}_{\boldsymbol{\theta}_t} + \boldsymbol{\delta}_t)$, where $\boldsymbol{g}_{\boldsymbol{\theta}} = \nabla_{\boldsymbol{\theta}}\mathcal{L}$, $\eta > 0$ is a constant step size and $\delta_t$ is the introduced collision.*

- *If the hard collision, which satisfies $\mathbb{E}[\boldsymbol{\delta}_t] = 0$ and $\mathbb{E}\|\boldsymbol{\delta}_t\|^2 \leq \sigma^2$ for all t, is applied, for any $\varepsilon > 0$, there exists a stepsize $\eta$ such that $\frac{1}{T}\sum_{t=1}^{T}\mathbb{E}[\|\boldsymbol{g}_{\boldsymbol{\theta}_t}\|^2] \leq \varepsilon$ after $\mathcal{O}(\frac{\sigma^2}{\varepsilon^2})$ iterations.*

- *If the soft collision, which satisfies $\|\boldsymbol{\delta}_t\| \leq D$ almost surely for all t, is applied, for any $\varepsilon > 0$, there exists a stepsize $\eta$ such that $\frac{1}{T}\sum_{t=1}^{T}\mathbb{E}[\|\boldsymbol{g}_{\boldsymbol{\theta}_t}\|^2] \leq \varepsilon$ after $\mathcal{O}(\frac{D^2}{\varepsilon^2})$ iterations.*

This theorem is proved with a traditional gradient descent optimizer as the base optimizer. Due to the boundedness of the collision term incorporated, we state that the convergence analysis on the

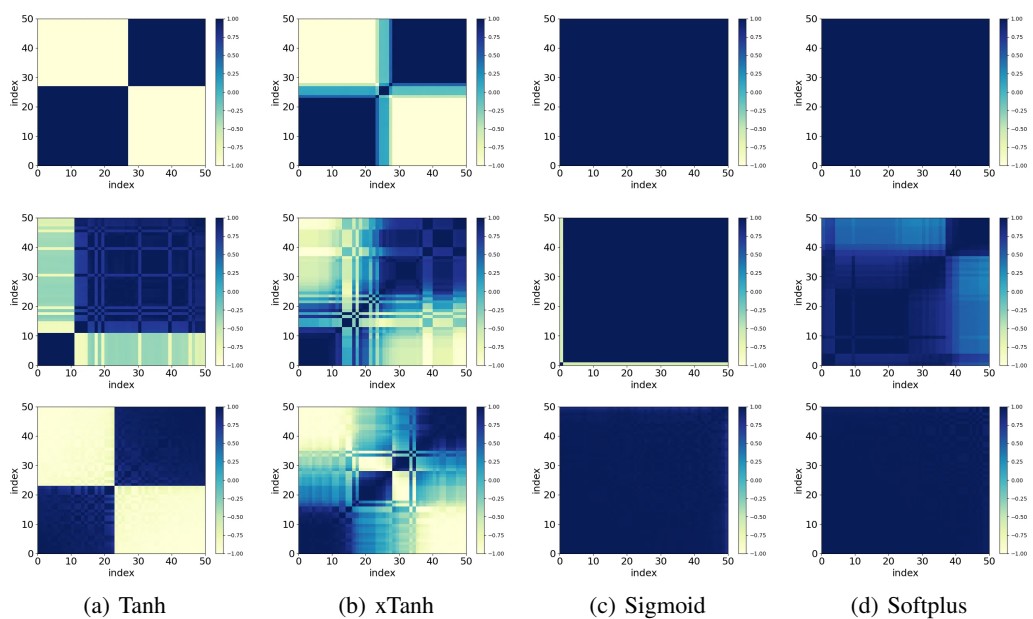

(a) Tanh      (b) xTanh      (c) Sigmoid      (d) Softplus

Figure 2: Condensation of two-layer NNs. The color indicates the cosine similarity of two hidden neurons' input weights at epoch 100, whose indexes are indicated by the abscissa and the ordinate, respectively. The activation functions are indicated by the sub-captions. The first row shows the model weights trained with the original Adam optimizer. The second and third rows depict the results with the Soft Collision and Hard Collision, respectively. A more detailed description on the plotting paradigm is included in Appendix B.1.

momentum, Adam-like or other types of gradient descent schema can be directly applied to the ones with the collision introduced. It will only incur a constant level change on the final result.

## 4 EXPERIMENTS

### 4.1 CONDENSATION EFFECTS STUDY

In this section, we validate the anti-condensation effects of the proposed models through experiments. We follow the experiment settings adopted by Zhou et al. (2022b). We use the cosine similarity of neuron weights to evaluate neuron similarity. The Adam optimizer (Kingma & Ba, 2017) is adopted as the base optimizer on both datasets. We visualize the cosine similarity matrix for better comparison. A more detailed description on the plotting paradigm is included in the Appendix.

**Experiments on a Multidimensional Synthetic Dataset** Firstly, we perform simple experiments on a multidimensional synthetic dataset. As in Zhou et al. (2022b), we utilize a 2-layer fully-connected network with a hidden dimension 5-50-1 to fit 80 instances sampled from a 5-dimensional function $\sum_{k=1}^{5} 3.5 \sin(5x_k + 1)$, where each $x_k$ is sampled uniformly from $[-4, 2]$. All parameters are initialized by a Gaussian distribution $\mathcal{N}(0, 0.005^2)$.

The result of the experiment is shown in Fig. 2. It could be seen that both the proposed hard collision and soft collision hinder the condensation of the weights. This result is in line with the mathematical proof and the physical interpretation. Further condensation experiments are shown in Appendix B.1. We could draw a simple conclusion in this section that the collision optimizers empirically halt the condensation in the network weights. We shall discuss its influence on model performance in the next section to test the practicality of our methods'.

### 4.2 EXPERIMENTS ON IMAGE AND TEXT DATASETS

**Experiments on CIFAR-10 and CIFAR-100.** Firstly, we perform experiments on two image datasets, CIFAR-10 and CIFAR-100 (Krizhevsky et al., 2009). ResNets of various sizes are chosen as the model backbone (He et al., 2015). A ResNet model can be separated into a convolutional encoder

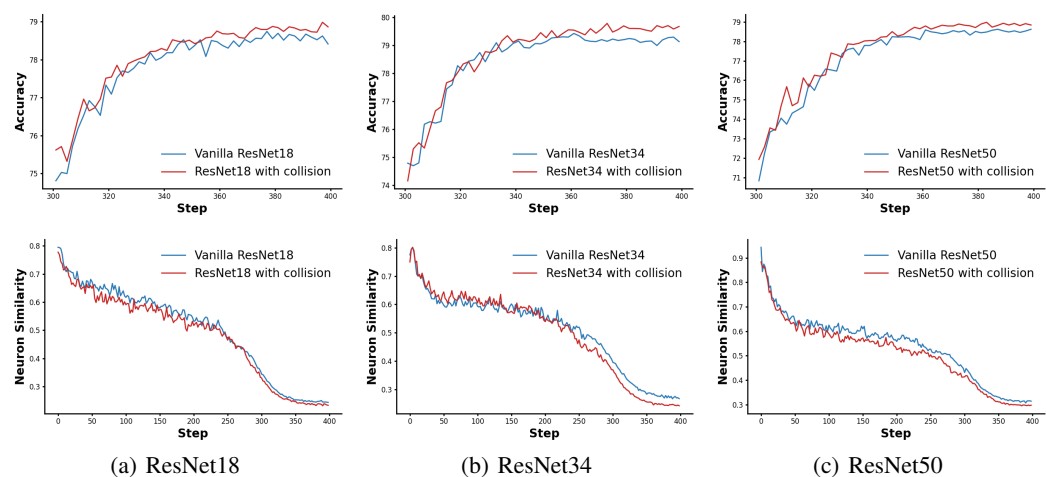

(a) ResNet18     (b) ResNet34     (c) ResNet50

Figure 3: Accuracy and neuron similarity of three different models on CIFAR-100. The upper row shows the test accuracy while the lower row depicts the neuron similarity changes during training.

and an FC-layer-based classifier. We apply our proposed collision mechanism to the classifier layer to avoid condensation in that layer.

The training process incorporates the SGD optimizer with a weight decay of $5 \times 10^{-4}$, a batch size of 128, a momentum of 0.9, and an initial learning rate of 0.1. A cosine annealing learning rate scheduler is further adopted to better control the learning process. All the models are trained for 200 epochs. Each image is padded by 4 pixels on each side, followed by random cropping of a $32 \times 32$ section from the padded image or its horizontal flip, as in the original paper. The prediction accuracy is chosen as the evaluation metric.

Table 1 presents results on CIFAR datasets. The introduction of the collision mechanism boosts the models' performance. Comparatively, the soft collision improves the test accuracy to a larger extent compared to the hard version. The soft collision approximately increases $0.5\%$ on the test accuracy, while the hard collision raises the accuracy by roughly $0.3\%$. Furthermore, the incorporation of collision enables shallower models to win over deeper models without collision. On CIFAR-10, ResNet18+Soft Collision predicts better than a vanilla ResNet50. This means that the introduction of this simple optimizer improvement is more effective than merely increasing network depth.

Table 1: Accuracy on CIFAR10 and CIFAR100. The collision is applied to the last fully-connected layer. S.C. is the abbreviation for Soft Collision, and H.C. is the abbreviation for Hard Collision.

| Model | CIFAR10 | CIFAR100 |
|---|---|---|
| ResNet18+SGD | $95.07\% \pm 0.18\%$ | $78.69\% \pm 0.38\%$ |
| ResNet18+S.C. | $\mathbf{95.74\% \pm 0.05\%}$ | $\mathbf{79.08\% \pm 0.12\%}$ |
| ResNet18+H.C. | $95.58\% \pm 0.07\%$ | $78.83\% \pm 0.09\%$ |
| ResNet34+SGD | $95.14\% \pm 0.37\%$ | $79.42\% \pm 0.45\%$ |
| ResNet34+S.C. | $\mathbf{95.76\% \pm 0.08\%}$ | $\mathbf{79.88\% \pm 0.25\%}$ |
| ResNet34+H.C. | $95.56\% \pm 0.14\%$ | $79.57\% \pm 0.32\%$ |
| ResNet50+SGD | $95.37\% \pm 0.38\%$ | $78.59\% \pm 0.56\%$ |
| ResNet50+S.C. | $\mathbf{95.83\% \pm 0.16\%}$ | $\mathbf{79.27\% \pm 0.23\%}$ |
| ResNet50+H.C. | $95.57\% \pm 0.05\%$ | $78.98\% \pm 0.18\%$ |

This improvement is notable since the collision mechanism basically introduces no extra time to the training and doesn't need to modify the model structure, the dataset (no extra augmentation is required), or the training schema. Specifically, it takes 3.68h to train a ResNet50 with the vanilla SGD optimizer and only 3.69h to train a ResNet50 with SGD and Soft Collision. The increase is negligible compared with the entire training duration.

To further show our methods' influence on the models, we plot the validation accuracy and neuron similarity changes during the training phase in Fig. 3. For simplicity, the neuron similarity is defined as the maximum abstract value in the cosine similarity matrix of the weight matrix. Though this neuron similarity cannot fully represent the similarity situation of the weight matrix, it serves as the upper bound of the similarity level and is still worth evaluating.

Fig.3 shows that our model can practically help to lower the weight similarity. The incorporation of collision helps to hinder condensation even on this authentic and commonly referred task. The collision mechanism helps the model to maintain a relatively smaller neuron similarity consistently

throughout the training. Note our method outperforms the vanilla ResNet model often in the last few epochs of training. It could be attributed to the results of the nature of the cosine annealing learning rate scheduler and the collision.

**Experiments on ImageNet-1K** In this section, we evaluate our model on a larger image classification dataset, ImageNet-1K (Deng et al., 2009). ImageNet-1K includes 1.28 million training images and 50,000 validation images across 1,000 categories. Following a recent paper (Wightman et al., 2021), we adopt a series of data augmentation and regularization strategies, including but not limited to RandAugment (Cubuk et al., 2019), Mixup alpha (Zhang et al., 2018), and CutMix alpha (Yun et al., 2019). The model training script is based on the timm library (Wightman, 2019).

Table 2: Experimental result on ImageNet-1K. All models are trained and validated at a resolution of $224 \times 224$. The collision is applied to the last fully-connected layer.

| Model | T-1 ACC.(%) | T-5 ACC.(%) |
|---|---|---|
| ResNet50+Lamb | 79.70% | 94.53% |
| ResNet50+S.C. | **80.23%** | **94.79%** |
| ResNet50+H.C. | 79.98% | 94.61% |
| ConvNext_Tiny+AdamW | 82.10% | 96.03% |
| ConvNext_Tiny+S.C. | **82.34%** | **96.07%** |
| ConvNext_Tiny+H.C. | 82.17% | 96.03% |
| ConvNext_Small+AdamW | 82.83% | 96.29% |
| ConvNext_Small+S.C. | **83.12%** | **96.33%** |
| ConvNext_Small+H.C. | 82.91% | 96.29% |

We adopt ResNet and ConvNext (Liu et al., 2022) as the training backbone. The ResNet 50 training hyperparameters are set as in Wightman et al. (2021). We follow the ConvNext hyperparameters as in Liu et al. (2024b). For both models, we still only apply collision to the last classifier layer. We evaluate both the Top-1 accuracy and Top-5 accuracy on the models.

Table 2 presents the results of various models on ImageNet-1K. In line with the results in CIFAR-10 and CIFAR-100, the introduction of collision, either the soft version or the hard version, boosts the model performance, both on the Top-1 accuracy and the Top-5 accuracy. The success for both ResNet models and ConvNext shows that our method is not restricted to the ResNet but can be applied to optimize various types of models.

**Experiments on IMDB and Snips.** In this section, we attempt to show that our method works not only on image tasks but also on text-related tasks. We choose two simple classification datasets, IMDB (Maas et al., 2011) and Snips (Coucke et al., 2018). IMDB comprises 25,000 highly polar movie reviews for training, and 25,000 for testing. Snips is a dataset of over 16,000 crowdsourced queries distributed among seven user intents for intent classification.

Table 3: Accuracy on the text classification dataset IMDB and Snips with pretrained models. The Soft and Hard Collisions are applied to train the fully-connected layer following the pretrained encoder.

| Pretrained Model | IMDB | Snips |
|---|---|---|
| bert-base-cased+AdamW | 92.52% | 98.14% |
| bert-base-cased+S.C. | **93.80%** | **98.43%** |
| bert-base-cased+H.C. | 93.42% | **98.43%** |
| bert-base-uncased+AdamW | 93.59% | 97.71% |
| bert-base-uncased+S.C. | **94.20%** | **97.89%** |
| bert-base-uncased+H.C. | 93.76% | 97.73% |
| distilbert-base-uncased+AdamW | 93.07% | 97.71% |
| distilbert-base-uncased+S.C. | **93.33%** | **98.43%** |
| distilbert-base-uncased+H.C. | 93.24% | 98.11% |

As with many text-based tasks, we train a sentiment classifier with a pretrained backbone. We mainly utilize BERT (Devlin et al., 2019) and DistilBert (Sanh et al., 2020) from the Transformers library by Hugging Face (Wolf et al., 2020). The classification model consists of a Bert Model transformer with a sequence classification head on top (a fully-connected layer). The collision is applied only to the classification head. Each classification model is trained for 10 epochs using an AdamW optimizer (Loshchilov & Hutter, 2019).

The results are shown in Table 3. On both of the datasets, collision improves the overall model performance. This result further proves that the collision mechanism works universally.

## 5 CONCLUSION AND LIMITATION DISCUSSION

We have presented KO, a novel neural optimizer gadget that interprets network parameter training through the perspective of kinetic theory. By modeling parameter updates as stochastic collisions in a particle system, KO introduces a physics-driven mechanism to tackle parameter condensation. KO is shown to be both physically and mathematically effective. Our framework uniquely bridges microscopic kinetic interactions with macroscopic learning dynamics, offering an alternative to conventional gradient-based optimization heuristics.

**Limitations:** computational overhead of simulating collision dynamics, which motivates future research on efficient approximations or hardware-accelerated implementations; extending KO to broader physical systems and analyzing its interplay with modern architectures.

ETHICS STATEMENT

This paper aims to propose a kinetics-inspired neural optimizer. While the research may entail various societal implications, we do not identify any that warrant specific emphasis in this paper.

REPRODUCIBILITY STATEMENT

All experimental results in the paper are reproducible, and the implementation code for reproducing experimental results will be fully open sourced on Github after the paper is accepted.

LLM USAGE STATEMENT

The contribution of LLM in the work proposed in this article is limited to: 1. polishing given written statements; 2. Given written sentence syntax review. We declare that no experimental data was generated/modified by LLM.

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

# A  PROOF

***Proof of Thm. 3.2***. Analyzing the changes in weight correlation of the network layer is equivalent to analyzing the cosine similarity matrix of the layer's weights. We term the weight matrix as $w$ and its i-th and j-th columns as $w_i$ and $w_j$ respectively. Their corresponding gradients are $g_i$ and $g_j$. After a step of gradient descent, we have:

$$w_i' = w_i - \eta(g_i + \Delta_i), w_j' = w_j - \eta(g_j + \Delta_j) \tag{9}$$

where $\Delta_i$ and $\Delta_j$ are the collision terms we apply to these two neurons, and $\eta$ is the learning rate. We will look further into the cosine similarity changes after the gradient descent to prove our methods' capacity. The weight cosine similarity can be calculated as:

$$C = \cos(w_i, w_j) = \frac{w_i^\top w_j}{\|w_i\|\|w_j\|}, C' = \frac{(w_i - \eta(g_i + \Delta_i))^\top (w_j - \eta(g_j + \Delta_j))}{\|w_i - \eta(g_i + \Delta_i)\|\|w_j - \eta(g_j + \Delta_j)\|} \tag{10}$$

We assume that the learning $\eta$ is a small number. Then we can use Taylor's expansion to get the following results:

$$C' \approx \frac{(w_i^\top w_j - \eta w_i^\top(g_j + \Delta_j) - \eta(g_i + \Delta_i)^\top w_j)(1 + \eta \frac{w_i^\top}{\|w_i\|^2}(g_i + \Delta_i))(1 + \eta \frac{w_j^\top}{\|w_j\|^2}(g_j + \Delta_j))}{\|w_i\|\|w_j\|}$$

$$= \frac{w_i^\top w_j(1 + \eta \frac{w_j^\top}{\|w_j\|^2}(g_j + \Delta_j) + \eta \frac{w_i^\top}{\|w_i\|^2}(g_i + \Delta_i)) - \eta w_i^\top(g_j + \Delta_j) - \eta(g_i + \Delta_i)^\top w_j}{\|w_i\|\|w_j\|}$$

$$= C + \eta \frac{w_i^\top w_j(\frac{w_j^\top}{\|w_j\|^2}(g_j + \Delta_j) + \frac{w_i^\top}{\|w_i\|^2}(g_i + \Delta_i)) - w_i^\top(g_j + \Delta_j) - (g_i + \Delta_i)^\top w_j}{\|w_i\|\|w_j\|}$$

$$= C + \eta \frac{w_i^\top w_j(\frac{w_j^\top}{\|w_j\|}g_j + \frac{w_i^\top}{\|w_i\|}g_i) - w_i^\top g_j - g_i^\top w_j}{\|w_i\|\|w_j\|}$$

$$+ \eta \frac{w_i^\top w_j(\frac{w_j^\top}{\|w_j\|^2}\Delta_j + \frac{w_i^\top}{\|w_i\|^2}\Delta_i) - w_i^\top \Delta_j - \Delta_i^\top w_j}{\|w_i\|\|w_j\|} \tag{11}$$

Next, we closely examine the last term in the above equation. The first two terms in Eq. 11 represent the original cosine similarity after the updates. We only need to prove that our collision can hinder condensation on top of the original pipelines.

**Soft Collision** We first examine the influence of the soft collision. We make a simple assumption that a collision only happens between $w_i$ and $w_j$. Then we can simplify the collision term to the following format:

$$\Delta_j = \alpha g_i, \Delta_i = \alpha g_j \tag{12}$$

where $\alpha$ is related to the similarity between neurons. We insert this into Eq. 11.

$$\delta C = w_i^\top w_j(\frac{w_j^\top}{\|w_j\|^2}\alpha g_i + \frac{w_i^\top}{\|w_i\|^2}\alpha g_j) - w_i^\top \alpha g_i - \alpha g_j^\top w_j$$

$$= \alpha \left[ (\cos(w_i, w_j)\frac{\|w_i\|}{\|w_j\|}w_j - w_i)^\top g_i + (\cos(w_i, w_j)\frac{\|w_j\|}{\|w_i\|}w_i - w_j)^\top g_j \right] \tag{13}$$

For simplicity, we focus on the training phase when it becomes stable; hence, we have $w_i^\top g_i < 0$ and $w_j^\top g_j < 0$. This is because when training is stabilized, $g_i$ won't push $w_i$ too harshly. We find that we can simplify the analysis of the first term in Eq. 13 to determine the sign of $\cos(w_i, w_j)\cos(w_j, g_i) - \cos(w_i, g_i)$ as $\|\frac{\|w_i\|}{\|w_j\|}w_j\| = \|w_i\|$. We make an assumption that $\theta(w_i, g_i) < \theta(w_j, g_i)$. The justification of this assumption is that $g_i$ is the gradient of $w_i$ and hence they are more correlated. All the analysis is based on the fact that we only analyze the dynamics in the stable training phase. In such a scenario, the angle between the weight vector $w_i$ and its corresponding gradient $g_i$ is expected to be small, indicating convergence toward a local minimum. Then $\cos(w_i, w_j)\cos(w_j, g_i) - \cos(w_i, g_i) < 0$. Since $\alpha$ is negatively correlated with

neuron similarity, then $\delta C$ pushes $C$ in the opposite direction of $C$ and hence lowers the overall cosine similarity. Conclusively, our soft collision helps to lower the overall weight correlation.

**Hard Collision** Next, we analyze how the hard collision functions. In the process of backpropagation, the negative gradient serves as the speed, and the weight matrix serves as the position. The hard collision procedure is detailed in the main paper. We still focus on the single collision between $w_i$ and $w_j$. The collision term is hence:

$$\Delta_i = \frac{g_j - g_i}{2} + \frac{1}{2}\|g_i - g_j\|u, \Delta_j = \frac{g_i - g_j}{2} - \frac{1}{2}\|g_i - g_j\|u \tag{14}$$

where $u$ is a random vector with no correlation with $w$. We combine this with Eq. 11:

$$\delta C = (\cos(w_i, w_j)\frac{\|w_i\|}{\|w_j\|}w_j - w_i - \cos(w_i, w_j)\frac{\|w_j\|}{\|w_i\|}w_i + w_j)^\top(\frac{g_i - g_j}{2} - \frac{1}{2}\|g_i - g_j\|u) \tag{15}$$

Since $u$ is a random vector, on average $-\frac{1}{2}\|g_i - g_j\|u$ will not bring any influence to $\delta C$. We only need to evaluate the terms that include $\frac{g_j - g_i}{2}$. Another important condition is that the collision only happens when $w_i$ and $w_j$ are close enough and have a large relative speed. This means that $(w_j - w_i)^\top(g_j - g_i) > 0$ and $\cos(w_i, w_j) > 0$. Since the original correlation is positive, we want a $\delta C < 0$ to decrease their weight similarity. Given the above assumption, we can turn to analyze the sign of $(w_j - w_i)^\top(g_i - g_j)$. Given that $(w_j - w_i)^\top(g_j - g_i) > 0$, we have $\delta C < 0$.

Now we have proven that both the soft and hard collision helps to reduce the weight correlation. The theorem gets proven. □

***Proof of Thm. 3.3.*** The proposed gradient updates with collision can be generalized in the form that $\theta' = \theta - \eta(g_\theta + \delta)$, where $\theta$ is the model parameters, $\delta$ refers to the introduced collision, $g_\theta$ represents the original model gradients and $\eta$ is the learning rate. We denote the loss function as $\mathcal{L}(\theta)$, and hence $g_\theta = \nabla_\theta \mathcal{L}$. Suppose $\mathcal{L}(\theta)$ is $L$-smooth, i.e. $\mathcal{L}(\theta') \leq \mathcal{L}(\theta) + g_\theta^\top(\theta' - \theta) + \frac{L}{2}\|\theta' - \theta\|^2$. If we apply our updates into this inequality, we will get:

$$\mathcal{L}(\theta') \leq \mathcal{L}(\theta) - \eta g_\theta^\top(g_\theta + \delta) + \frac{L\eta^2}{2}\|g_\theta + \delta\|^2 \tag{16}$$

**Hard Collision** We will first prove the convergence for hard collision. We summarize several properties of the proposed collision mechanism. Firstly, our collision is a zero-mean collision as indicated from the derivation formula. Hence, we have $\mathbb{E}(\delta_t) = 0$. Secondly, we assume that the gradients are bounded, which is commonly adopted in previous literature Bottou et al. (2018). Since our proposed collision term is built upon the gradients as shown in Eq. 6, we can easily proved that $\delta$ is also bounded. Since bounded random variables always have a finite variance, then there exists a $\sigma$ that bounds $\delta$'s variance, i.e. $\mathbb{E}[\|\delta\|^2] \leq \sigma^2$.

We would love to investigate the average performance of this stochastic algorithm. We expand Eq. 16 and take the expectation and get:

$$\mathbb{E}[\mathcal{L}(\theta')] \leq \mathcal{L}(\theta) - \eta(1 - \frac{L\eta}{2})\|g_\theta\|^2 - \eta(1 - L\eta)g_\theta^T\mathbb{E}[\delta] + \frac{L\eta^2}{2}\mathbb{E}[\|\delta\|^2]$$
$$\leq \mathcal{L}(\theta) - \eta(1 - \frac{L\eta}{2})\|g_\theta\|^2 + \frac{L\eta^2}{2}\sigma^2 \tag{17}$$

We can pick an $\eta_t > 0$ for this iteration that satisfies $\eta_t(1 - \frac{L\eta_t}{2}) > 0$. Next, we want to prove the convergence. We sum Eq. 17 of various epochs from 1 to T, we have:

$$\sum_{t=1}^{T} \eta_t(1 - \frac{L\eta_t}{2})\mathbb{E}[\|g_{\theta_t}\|^2] \leq \mathcal{L}(\theta_0) - \mathbb{E}[\mathcal{L}(\theta_{T+1})] + \sum_{t=1}^{T} \frac{L\eta_t^2}{2}\sigma^2 \tag{18}$$

If we take $\eta_t < \frac{1}{L}$, then $1 - \frac{L\eta_t}{2} > \frac{1}{2}$. Also, since our aim is to minimize $\mathcal{L}(\theta)$, it is reasonable for us to assume that $\mathcal{L}(\theta)$ has a lower bound. Under constant step sizes, we can have that:

$$\frac{1}{T}\sum_{t=1}^{T} \mathbb{E}[\|g_{\theta_t}\|^2] \leq \frac{2}{\eta T}(\mathcal{L}(\theta_0) - \mathbb{E}[\mathcal{L}(\theta_{T+1})]) + L\eta\sigma^2 \tag{19}$$

If we set $\eta = \frac{\varepsilon}{2L\sigma^2}$, then $L\eta\sigma^2 \leq \frac{\varepsilon}{2}$. If we further require $T \geq \frac{4(\mathcal{L}(\boldsymbol{\theta}_0) - \mathbb{E}[\mathcal{L}(\boldsymbol{\theta}_{T+1})])}{\eta\varepsilon} = \mathcal{O}(\frac{\sigma^2}{\varepsilon^2})$, we can achieve $\frac{1}{T}\sum_{t=1}^{T}\mathbb{E}[\|\boldsymbol{g}_{\boldsymbol{\theta}_t}\|^2] \leq \varepsilon$.

**Soft Collision** As for the soft collision, we can prove it by assuming that $\boldsymbol{\delta}$ is uniformly bounded by $D$. This assumption can also be justified by the bound assumption on the gradients since the cosine similarity is bounded by 1. Using this property we have:

$$|\boldsymbol{g}_{\boldsymbol{\theta}}(\boldsymbol{\theta})^\top \boldsymbol{\delta}| \leq \|\boldsymbol{g}_{\boldsymbol{\theta}}(\boldsymbol{\theta})\|\|\boldsymbol{\delta}\| \leq \|\boldsymbol{g}_{\boldsymbol{\theta}}(\boldsymbol{\theta})\|D \tag{20}$$

If we insert this into Eq. 16, we have:

$$\mathcal{L}(\boldsymbol{\theta}') \leq \mathcal{L}(\boldsymbol{\theta}) - \eta(1 - \frac{L\eta}{2})\|\boldsymbol{g}_{\boldsymbol{\theta}}\|^2 + (L\eta^2 - \eta)\|\boldsymbol{g}_{\boldsymbol{\theta}}\|D + \frac{L\eta^2}{2}D^2 \tag{21}$$

Then, we want to get rid of the term with solely $\|\boldsymbol{g}_{\boldsymbol{\theta}}\|$. If we apply Young's inequality on this, we get:

$$\|\boldsymbol{g}_{\boldsymbol{\theta}}\|D \leq \frac{\alpha}{2}\|\boldsymbol{g}_{\boldsymbol{\theta}}\|^2 + \frac{1}{2\alpha}D^2 \tag{22}$$

where $\alpha > 0$ is a constant. For the choice of $\eta$, we can always find an upper bound of it because of our diminishing step-size training scheme. We denote the upper bound as $\hat{\eta}$. Furthermore, if we choose $\eta$s which follows $L\eta - 2 > 0$, we can formulate our inequality as:

$$\begin{aligned}
\mathcal{L}(\boldsymbol{\theta}') &\leq \mathcal{L}(\boldsymbol{\theta}) - \eta(1 - \frac{L\eta}{2})\|\boldsymbol{g}_{\boldsymbol{\theta}}\|^2 + (L\eta^2 - \eta)(\frac{\alpha}{2}\|\boldsymbol{g}_{\boldsymbol{\theta}}\|^2 + \frac{1}{2\alpha}D^2) + \frac{L\eta^2}{2}D^2 \\
&= \mathcal{L}(\boldsymbol{\theta}) - \eta(1 - \frac{L\eta}{2} + (1 - L\eta)\frac{\alpha}{2})\|\boldsymbol{g}_{\boldsymbol{\theta}}\|^2 + (\frac{L\eta^2}{2} + (L\eta^2 - \eta)\frac{1}{2\alpha})D^2 \\
&\leq \mathcal{L}(\boldsymbol{\theta}) - \eta(1 - \frac{L\hat{\eta}}{2} + (1 - L\hat{\eta})\frac{\alpha}{2})\|\boldsymbol{g}_{\boldsymbol{\theta}}\|^2 + \eta^2(\frac{L}{2} + \frac{L}{2\alpha})D^2 \\
&= \mathcal{L}(\boldsymbol{\theta}) - c\eta\|\boldsymbol{g}_{\boldsymbol{\theta}}\|^2 + c'\eta^2 D^2
\end{aligned} \tag{23}$$

where $c, c'$ are two constants independent of t. Then, as in the previous case, we sum from epoch 1 to T, we get:

$$\frac{1}{T}\sum_{t=1}^{T}\mathbb{E}[\|\boldsymbol{g}_{\boldsymbol{\theta}_t}\|^2] \leq \frac{2}{c\eta T}(\mathcal{L}(\boldsymbol{\theta}_0) - \mathbb{E}[\mathcal{L}(\boldsymbol{\theta}_{T+1})]) + \frac{c'\eta D^2}{c} \tag{24}$$

Firstly, we choose $\eta = \frac{c\varepsilon}{2c'D^2}$, then $\frac{c'\eta D^2}{c} \leq \frac{\varepsilon}{2}$. If we further let $T \geq \frac{4(\mathcal{L}(\boldsymbol{\theta}_0) - \mathbb{E}[\mathcal{L}(\boldsymbol{\theta}_{T+1})])}{c\eta} = \mathcal{O}(\frac{D^2}{\varepsilon^2})$, we can get $\frac{1}{T}\sum_{t=1}^{T}\mathbb{E}[\|\boldsymbol{g}_{\boldsymbol{\theta}_t}\|^2] \leq \varepsilon$.

Furthermore, we state that the analysis on the momentum, Adam-like or other types of gradient descent schema can be directly applied to the ones with the collision mechanism. This is due to the boundedness of the collision term. It will only incur a constant level change on the final result. $\square$

# B  ADDITIONAL EXPERIMENTS AND RESULTS

## B.1  CONDENSATION EXPERIMENTS ON CIFAR-10.

Firstly, we detail the plotting scheme used for condensation visualization. To be specific, after we get the cosine similarity matrix for the weight matrix, we perform clustering based on this similarity. Starting from neuron 0, we iteratively groups neurons that are strongly similar (>0.6) to the current neuron. If a negatively correlated neuron (< -0.6) is found, it may become the new reference. After we get this clusters of neurons, we rearranges the cosine similarity matrix based on the new order. The visualized picture represents different clusters of neurons. When the reordered cosine similarity matrix shows large, dense blocks, it indicates that many neurons are highly correlated and thus condensed into a small number of clusters. In contrast, when the similarity matrix exhibits more fragmented or evenly distributed clusters, it implies weaker correlations among neurons.

Secondly, we test our methods' anti-condensation effects on a more practical example, CIFAR-10 (Krizhevsky et al., 2009). We follow the ResNet18-like structure introduced in Zhou et al. (2022b): the original single fully-connected (FC) layer is replaced with a series of FC layers of size 1024-1024-10. The results are visualized in Fig. 4. The learning rate is $3 \times 10^{-8}$ for Tanh activation and $5 \times 10^{-6}$ for xTanh activation. As in the synthetic data experiments, both of the proposed collision methods slow down the original condensation.

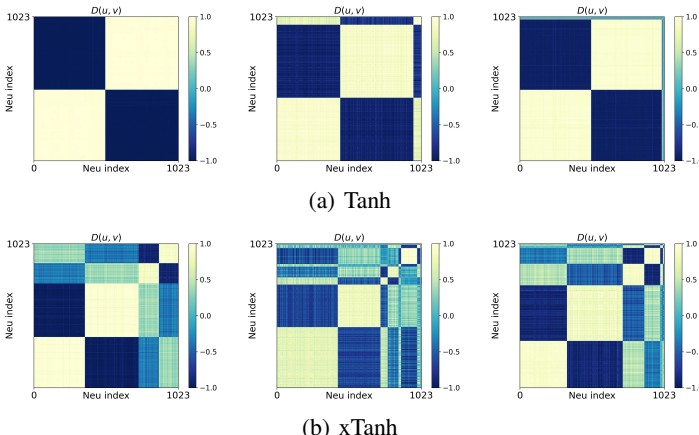

(a) Tanh

(b) xTanh

Figure 4: Condensation of Resnet18-like neural networks on CIFAR-10. The color in the figures indicates the cosine similarity of the normalized input weights of two neurons in the first FC layer. The subcaption represents the activation function in the FC layers. The first column shows the model weights trained with the original Adam optimizer. The second and third columns depict the results with the Soft Collision and Hard Collision, respectively.

## B.2 MEMORY AND RUNTIME

In this section we would love to discuss the memory and runtime issues possibly incurred by the collision mechanism. The results are shown in Table 4 and Table 5. We can infer from the table that the soft collision mechanism incurs minimal additional memory consumption. This is because it avoids the construction of large auxiliary matrices, making it suitable even for relatively large models. In contrast, the hard collision method involves the computation of a relative speed matrix, which introduces additional memory overhead. While this cost is manageable for small to medium-scale networks, it may become prohibitive for very large models with a high number of neurons (particles), thus limiting its practical applicability in such settings.

That being said, we emphasize that soft collision not only reduces memory usage but also achieves superior empirical performance. However, we include results for hard collision because, despite its higher memory consumption, it offers a more direct and interpretable connection to the physical dynamics of the collision. In fact, the soft collision mechanism was originally inspired by the hard version, aiming to retain key conceptual benefits while improving efficiency.

|  | RunTime | Memory |
|---|---|---|
| Vanilla Resnet18 | 5523s | 2120M |
| Resnet18+S.C. | 5791s | 2120M |
| Resnet18+H.C. | 8784s | 2124M |
| Vanilla Resnet34 | 5915s | 2612M |
| Resnet34+S.C. | 6481s | 2612M |
| Resnet34+H.C. | 7293s | 2616M |
| Vanilla Resnet50 | 12039s | 5172M |
| Resnet50+S.C. | 14871s | 5172M |
| Resnet50+H.C. | 17334s | 5176M |

Table 4: Runtime (in seconds) and memory (in MB) for different ResNet architectures with Vanilla, soft collision, and hard collision on CIFAR-10.

|  | RunTime | memory |
|---|---|---|
| Vanilla Resnet18 | 5774s | 2122M |
| Resnet18+S.C | 5802s | 2122M |
| Resnet18+H.C | 16389s | 2126M |
| Vanilla Resnet34 | 8857s | 2612M |
| Resnet34+S.C | 8968s | 2612M |
| Resnet34+H.C | 18405s | 2616M |
| Vanilla Resnet50 | 13308s | 11576M |
| Resnet50+S.C | 13310s | 5176M |
| Resnet50+H.C | 26438s | 11576M |

Table 5: Runtime (in seconds) and memory (in MB) for different ResNet architectures with Vanilla, soft collision, and hard collision on CIFAR-100.

## B.3 LLM EXPERIMENT

We conducted additional pre-training experiments on Large Language Models (LLMs). Specifically, we trained Qwen2-0.5B and Qwen2-1.5B models on the FineWeb-Edu dataset for 50B and 25B tokens, respectively. Our proposed soft collision method (S.C.) was applied to the last transformer layer. We evaluated the models on a diverse set of challenging benchmarks, including Knowledge-intensive tasks (ARC-C, ARC-E) and Commonsense Reasoning (HellaSwag, WinoGrande, OBQA, PIQA).

As shown in Table 6, our method consistently outperforms the baseline. We achieve an average absolute improvement of $> 1.4\%$ across all tasks (e.g., $+1.6\%$ on Qwen2-1.5B). Crucially, this performance gain is approximately equivalent to a 10% reduction in training steps to reach baseline accuracy. This benefit significantly outweighs the minor computational overhead ($\sim 3.3\%$) introduced by our method. This provides strong evidence that our approach offers a favorable trade-off between training cost and model performance.

Table 6: Comparison of model performance on various benchmarks.

| Model \ Acc ↑ | PIQA | ARC_C | ARC_E | HellaSwag | WinoGrande | OBQA | Avg |
|---|---|---|---|---|---|---|---|
| Qwen2-0.5B | 0.575 | 0.227 | 0.371 | 0.273 | 0.498 | 0.246 | 0.365 |
| Qwen2-0.5B-S.C. | **0.582** | **0.241** | **0.385** | **0.276** | **0.522** | **0.268** | **0.379** |
| Qwen2-1.5B | 0.602 | 0.236 | 0.439 | 0.310 | 0.495 | 0.256 | 0.390 |
| Qwen2-1.5B-S.C. | **0.615** | **0.258** | **0.450** | **0.313** | **0.535** | **0.266** | **0.406** |

