# OpenReview forum: "KO: Kinetics-inspired Neural Optimizer with PDE Simulation Approaches"
_ICLR.cc/2026/Conference — Submitted to ICLR 2026_

### Official Review · Reviewer_Xj7q · 2025-10-31

**Soundness:** 2
**Presentation:** 2
**Contribution:** 2
**Rating:** 4
**Confidence:** 3

**Summary:**

The paper presents a method to improve optimization of neural network trough a combination of the gradient with a mechanism that encourages neurons to learn different features. This mechanism is inspired by the kinetic theory of ideal gases from physics. Experiments were conducted on a synthetic problem, image and text datasets.

**Strengths:**

1) The authors present an interesting connection of neural network optimization to kinetic gas theory.

2) The authors present a small example on a synthetic dataset, which attempts to give some insights and illustration of their method beyond the usual evaluation on large-scale tasks.

3) The paper states that there are improvements in the experiments through their method. However, I am unsure how valuable improvements of ~0.5% in classification accuracy are to computer vision or NLP communities.

**Weaknesses:**

1) For me, the paper was not easy to read and I think I would not be able to implement the method from the paper.
- More formal definitions of terms like "parameter diversity", "parameter condensation". "distribution of network parameters" (What are the different samples? Parameters of neurons, of layers?)
- Often the phrasing is vague "We introduce collisions at the gradient level" (L190)
- What's often helpful to explain analogies is a table that makes the connections explicit (e.g. weight vector of one neuron = particle)


2) Some practical aspects should be discussed in more detail:
- Does the method require additional hyperparameters? Do they need additional tuning?
- How does the computational cost scale with the number of parameters?

3) Limited experimental evaluation: The authors should compare their method to other methods that also aim to manipulate the weights to improve generalization, such as weight decay.

4) The goal of avoiding similar features in neurons to improve generalization, which is the basis of this work, appears to be in contradiction to more established views on generalization like the one in [1]. Can you discuss if and to what extent this is indeed true and what shortcomings you see in the perspective in [1]?

[1] R. S. Sutton, A. G. Barto "Reinforcement Learning, An Introduction" (Section 9.5.3 and Figure 9.7 Generalization in linear function approximation...)

**Questions:**

- See also my questions in "weaknesses"
- Why was the network on the synthetic dataset initialized with weights near zero? Does the experiment still produce the same results with the usual initilization schemes for gradient-based neural network optimization?
- Please fix equation references in L215

---

> ### Author Response · Authors · 2025-11-25
>
> **Q1: More formal definitions; clarification of value phrasing; a table explain connection between weight vector of one neuron = particle**
>
> A1: We thank the reviewer for highlighting these points. We are happy to provide formal definitions and clarify the analogy between the physical system and our optimization framework below.
>
> 1. **Formal Definitions**
> In our framework, **parameter condensation** and **parameter diversity** are inversely related and are quantitatively defined using the average cosine similarity of the weight vectors.
> Let $W \in \mathbb{R}^{D \times N}$ denote the weight matrix of a layer, where $N$ is the number of neurons (particles) and $D$ is the dimension of the weight weights. We denote the $j$-th column of $W$ as $w_j$.
> *   **Parameter Condensation:** We define the degree of condensation, $\rho(W)$, as the average absolute cosine similarity between all pairs of distinct column vectors:
>     $$
>     \rho(W) = \frac{1}{N(N-1)} \sum_{i \neq j} \frac{|w_i^\top w_j|}{\|w_i\|_2 \|w_j\|_2}
>     $$
>     A higher $\rho(W)$ indicates that neurons are collapsing into a low-dimensional subspace (high condensation).
> *   **Parameter Diversity:** This is conceptually the inverse of condensation. A lower $\rho(W)$ corresponds to greater angular separation between weight vectors, indicating higher parameter diversity.
> *   **Parameter Distribution:** We treat the collection of weight vectors $\{w_1, \dots, w_N\}$ as an empirical distribution of $N$ particles in the spatial domain $\mathbb{R}^D$. The optimization process is then viewed as the evolution of this particle system over time.
>
> 2. **Clarification of Phrasing**
> We apologize for any ambiguity in the initial text. To clarify: we map the **parameter update rule** of the optimizer to the **motion of particles** in a physical system.
> *   Motivation: Empirically, excessive similarity (condensation) among neurons harms generalization. In kinetic theory, particles maintain separation through collisions.
> *   Mechanism: We introduce "collisions" at the **gradient level**. In our analogy, if weights represent positions, then gradients (or negative gradients) represent velocities. When neurons become too similar (i.e., particles get too close), our method introduces a repulsive term (simulating a collision) to the gradients. This modifies the "velocity" of the parameters, pushing them apart to maintain diversity without altering the fundamental loss minimization objective.
>
> 3. Connection between Physics and Neural Networks
> We have summarized the correspondence between the concepts in Kinetic Theory and Neural Network Optimization in the table below:
>
> | **Concept** | **Kinetic Theory (Physical System)** | **Neural Network Optimization** |
> | :--- | :--- | :--- |
> | **Agent** | A single particle | A single neuron (represented by its weight vector $w_j$) |
> | **Position** | Spatial coordinates of the particle | Values of the weight vector $w_j \in \mathbb{R}^D$ |
> | **Velocity** | Particle velocity vector | Gradient of the weight vector (update direction) $g_j$ |
> | **Interaction** | Stochastic collisions | Gradient modification (Hard/Soft Collision terms) |
> | **System Dynamics** | Boltzmann Transport Equation | Optimization trajectory (Gradient Descent + Collision) |
>
> ***
>
> **Q2: Does the method require additional hyperparameters? Do they need additional tuning?**
>
> A2:Thank you for this important question regarding practical implementation.
>
> 1. Additional Hyperparameters. Yes, KO introduces a single hyperparameter, coll_coef (collision coefficient), which governs the intensity or probability of the collision mechanism. This parameter determines the strength of the repulsive force applied to prevent parameter condensation.
>
> 2. Sensitivity and Tuning. While coll_coef can be tuned to achieve peak performance for specific architectures, our method does not require extensive tuning to yield improvements.
>
> To demonstrate this, we performed a sensitivity analysis on a ResNet-50 model trained on CIFAR-10 (see the table below). We varied coll_coef from 0.1 to 0.9. The results show that KO consistently outperforms the baseline (coll_coef = 0.0) across the entire range of values.
>
> This indicates that the proposed method is highly robust. Practitioners can select a default value (e.g., 0.5) and expect performance gains without a burdensome hyperparameter search.
>
> | coll_coef | 0.0 | 0.1 | 0.2 | 0.3 | 0.4 | 0.5 | 0.6 | 0.7 | 0.8 | 0.9 |
> | --------- | --- | --- | --- | --- | --- | --- | --- | --- | --- | --- |
> |ResNet50+S.C.  Acc|77.38%|78.23%|78.80%|78.30%|77.98%|**78.92%**|77.52%|77.87%|78.01%|78.16%|

---

> > ### Author Response · Authors · 2025-11-25
> >
> > **Q3: How does the computational cost scale with the number of parameters?**
> >
> > A3: We thank the reviewer for this insightful question regarding scalability. For simplicity, we assume the collision is applied to the linear layer with weight matrix $W \in \mathbb{R}^{n \times n}$ and gradients of the same size, and the analysis depends on the interaction dimension $n$.
> > 1. **Soft Collision: Compute-Bound and Efficient**
> > For soft collision (collision type wg), the algorithm primarily involves matrix multiplications to compute cosine similarity matrices (e.g., $W W^T$) and the repulsion force.
> > - Time Complexity: The operation is dominated by matrix multiplication, resulting in a theoretical time complexity of $O(n^3)$.
> > - Space Complexity: It requires storing intermediate similarity matrices, scaling with $O(n^2)$.
> > - Practical Performance: Although $O(n^3)$ might seem computationally heavy, it translates to high arithmetic intensity operations (pure matrix multiplications) that are heavily optimized on modern GPUs (e.g., via Tensor Cores). Consequently, the actual runtime overhead is negligible compared to the memory-bound operations of a full training step.
> > 2. **Hard Collision: Memory-Bound and Expensive**
> > In contrast, the hard collision mechanism simulates physical N-body interactions, which requires broadcasting tensors to compute pairwise relative positions and velocities. This broadcasting creates intermediate tensors of size $O(n^3)$, causing the space complexity to scale cubically. This makes hard collision a memory-bound process, significantly increasing the risk of memory overflow (OOM) and reducing training speed due to memory bandwidth limitations.
> > 3. Empirical Validation:
> > We have provided detailed statistics on training time and memory consumption for both mechanisms in Appendix B.2 of the draft. As shown in the appendix, soft collision introduces minimal overhead (consistent with our GPU efficiency analysis), while hard collision shows a noticeable increase in resource usage, validating our complexity analysis.
> > 4. Summary:
> > Soft collision scales efficiently even for large architectures due to its compatibility with GPU hardware acceleration. Hard collision, while effective, incurs a heavier memory footprint ($O(n^3)$ space complexity), making it less suitable for extremely large layers.
> >
> > **Q4: The authors should compare their method to other methods that also aim to manipulate the weights to improve generalization, such as weight decay.**
> >
> > A4: Thank you for this important suggestion. First, we would like to clarify that KO is designed as a "plug-and-play" module compatible with standard optimizers. Consequently, **all results reported in our main manuscript already include weight decay** as part of the standard training recipe for CIFAR-10/100. The performance gains shown in the paper are therefore in addition to the benefits of weight decay.
> >
> > To explicitly isolate the contribution of KO versus weight decay, we provide an ablation study below. The results demonstrate three key findings:
> >
> > 1. **KO outperforms Weight Decay**: Models trained with KO alone (e.g., column S.C) consistently outperform those trained with Weight Decay alone (Vanilla + WD).
> > 2. **Complementary Benefits**: When KO is added to a baseline that already uses Weight Decay, performance improves further (e.g., ResNet34 improves from 95.14% to 95.76%).
> > 3. **Compatibility**: Our method functions effectively both with and without weight decay, confirming that KO targets a different aspect of generalization (reducing neuron condensation) than L2 regularization.
> >
> > |          | Vanilla      | Vanilla + WeightDecay | S.C          | S.C + WeightDecay     | H.C          | H.C + WeightDecay       |
> > | -------- | ------------ | ------------ | ------------ | ------------ | ------------ | -------------- |
> > | Resnet18 | 95.02% $\pm$ 0.29% | 95.07% $\pm$ 0.18% | 95.74% $\pm$ 0.09% | 95.74% $\pm$ 0.05% | 95.42% $\pm$ 0.13% | 95.58% $\pm$ 0.07% |
> > | Resnet34 | 95.11% $\pm$ 0.32%| 95.14% $\pm$ 0.37% |95.52% $\pm$ 0.12%| 95.76% $\pm$ 0.08% |95.33% $\pm$ 0.18%| 95.56% $\pm$ 0.14% |
> > | Resnet50 | 95.31% $\pm$ 0.27% |95.37% $\pm$ 0.38% |95.62% $\pm$ 0.08%| 95.83% $\pm$ 0.16% |95.42% $\pm$ 0.11%| 95.57% $\pm$ 0.05% |

---

> > > ### Author Response · Authors · 2025-11-25
> > >
> > > **Q5: The goal of avoiding similar features in neurons to improve generalization, which is the basis of this work, appears to be in contradiction to more established views on generalization, like the one in [1]. Can you discuss if and to what extent this is indeed true and what shortcomings you see in the perspective in [1]?**
> > >
> > > A5: Thank you for this insightful question. We believe there is a misunderstanding regarding the specific "similarity" we address. It is crucial to distinguish between Feature Similarity (output representations) and Weight/Neuron Similarity (internal parameters).
> > >
> > > 1. **Clarification**: The perspective in [1] and established views correctly argue that diverse features improve generalization. Our work does not contradict this. Instead, we focus on the weights of neurons. Weight condensation refers to neurons having identical or highly correlated weight vectors, which is an issue in the parameter space, distinct from feature similarity in the activation space.
> > > 2. **Connection**: Far from contradicting [1], our method complements it. Mathematically, weight condensation reduces the rank of the weight matrices. A low-rank projection limits the network's capacity to generate diverse features. By preventing weight condensation (i.e., maintaining higher rank weights), our method preserves the necessary conditions for generating diverse features.
> > >
> > > Therefore, our approach can be viewed as a fundamental intervention on the parameters that supports the generation of distinct features, aligning with rather than opposing the generalization goals in [1].
> > >
> > > [1] R. S. Sutton, A. G. Barto "Reinforcement Learning, An Introduction" (Section 9.5.3 and Figure 9.7 Generalization in linear function approximation...)
> > >
> > > **Q6: Why was the network on the synthetic dataset initialized with weights near zero? Does the experiment still produce the same results with the usual initialization schemes for gradient-based neural network optimization?**
> > >
> > > A6: We appreciate this opportunity to clarify our experimental design.
> > > 1. **Motivation for near-zero initialization**: We employed near-zero initialization on the synthetic dataset primarily to amplify the "condensation" phenomenon described in [2]. In this setting, weights tend to condense significantly, making the phenomenon visually distinct. This setup serves as a "magnifying glass," allowing us to clearly visualize the problem and intuitively evaluate whether our proposed method effectively mitigates it.
> > > 2. **Standard initialization**: Yes, our findings hold under standard settings. While condensation is less visually extreme with standard initialization schemes, it persists and continues to hinder generalization. As demonstrated in Figure 3 (using ResNet18/34/50), our method effectively alleviates condensation and improves performance under standard initialization and training protocols.
> > >
> > > Thus, the synthetic setup reveals the mechanism, while the ResNet experiments confirm its practical efficacy.
> > >
> > > [2] Luo, T., Xu, Z. Q. J., Ma, Z., & Zhang, Y. (2021). Phase diagram for two-layer relu neural networks at infinite-width limit. Journal of Machine Learning Research, 22(71), 1-47.

---

> > > > ### Author Response · Authors · 2025-11-25
> > > >
> > > > **Q7: However, I am unsure how valuable improvements of ~0.5% in classification accuracy are to computer vision or NLP communities.**
> > > >
> > > > A7: We appreciate the reviewer’s comment regarding the significance of performance gains. While a ~0.5% improvement may appear marginal in some classification settings, we argue that the efficiency implications of such gains are substantial. To demonstrate this scalability and value, we conducted additional pre-training experiments on Large Language Models (LLMs).
> > > >
> > > > Specifically, we trained Qwen2-0.5B and Qwen2-1.5B models on the FineWeb-Edu dataset for 50B and 25B tokens, respectively. Our proposed soft collision method (S.C.) was applied to the last transformer layer. We evaluated the models on a diverse set of challenging benchmarks, including Knowledge-intensive tasks (ARC-C, ARC-E) and Commonsense Reasoning (HellaSwag, WinoGrande, OBQA, PIQA).
> > > >
> > > > As shown in the table below, our method consistently outperforms the baseline. We achieve an average absolute improvement of >1.4% across all tasks (e.g., +1.6% on Qwen2-1.5B). Crucially, this performance gain is approximately equivalent to a **10% reduction in training steps** to reach baseline accuracy. This benefit significantly outweighs the minor computational overhead (~3.3%) introduced by our method.
> > > > This provides strong evidence that our approach offers a favorable trade-off between training cost and model performance.
> > > >
> > > >
> > > >
> > > >
> > > > | Model \ Acc $\uparrow$ | PIQA      |ARC\_C   | ARC\_E   | HellaSwag | WinoGrande | OBQA | Avg |
> > > > | -------------------- | --------- |---------| ---------| ---- | ---- | ---- | ---- |
> > > > | Qwen2-0.5B                 | 0.575     |0.227    | 0.371   |0.273 |0.498 |0.246 | 0.365 |
> > > > | Qwen2-0.5B-S.C.           | **0.582** |**0.241**|**0.385**|**0.276**|**0.522**| **0.268**    | **0.379** |
> > > > | Qwen2-1.5B                 | 0.602     |0.236    | 0.439   |0.310 |0.495 |0.256 | 0.390 |
> > > > | Qwen2-1.5B-S.C.           | **0.615** |**0.258**|**0.450**|**0.313**|**0.535**| **0.266**    | **0.406** |

---

### Official Review · Reviewer_uF9q · 2025-10-31

**Soundness:** 3
**Presentation:** 3
**Contribution:** 3
**Rating:** 8
**Confidence:** 4

**Summary:**

The paper introduces a new method for training the weights of neural network architectures for tasks like image classification. The method aims to address the issue that standard gradient based methods can lead to layers with neurons in the same lower dimensional subspaces, and this leads to worse model generalization. The training method works by adjusting the gradient following an approach motivated by physics: particle collisions. Theoretical and numerical results support the finidings.

**Strengths:**

- The paper introduces a new training method that adjusts the gradients during updates, using ideas inspired from physics - particle collisions. The method is quite agnostic to the rest of the training process and can be employed in many training frameworks and optimizers.

- Numerically, the results are very convincing in improving the performance of other optimizers.

- Theory justifies these results - reducing the layer weight correlation.

- Computationally the method is not heavy.

**Weaknesses:**

- In terms of comparisons, I would have expected that the authors compare more clearly with regularization based approaches. In the introduction, the latter are described as post-hoc, but I am not convinced about this. In principle, regularization addresses the same issue, poor generalization.

- Convergence is proven, but it is not clear what is the impact of the proposed approach. I expect that the introduced 'collisions' cause the gradient directions to change randomly, and as a result, creating more variance in some cases?

**Questions:**

- I wonder how the 'collision' setup should change across the depth of the neural network, as well as during training. This aspect is not clear. Should more collisions happen earlier during training, or in earlier layers?

- The method is argued to increase the entropy of the weights from one step to the next one. How does this compare with entropy being used as a regularization term? This relates to my comment earlier about unclear comparison with regularization based approaches

- The improvements are higher in less complex classification problems and also smaller neural networks. How is this justified?

- Are the obtained networks more susceptible to adversarial perturbations? Intuitively, maybe yes, since they have more entropy/randomness/high dimensional subspaces. This would be nice to show but not strictly a drawback

---

> ### Author Response · Authors · 2025-11-25
>
> **Q1: Convergence is proven, but it is not clear what the impact of the proposed approach is. I expect that the introduced 'collisions' cause the gradient directions to change randomly, and as a result, create more variance in some cases.**
>
> A1: Thank you for your insightful question. We appreciate the opportunity to clarify the distinction between our kinetics-inspired approach and random gradient noise.
>
> 1.  **Structured Repulsion vs. Random Noise:** While our "Hard Collision" module employs stochastic sampling (inspired by the DSMC method), it is fundamentally different from adding isotropic noise (e.g., Gaussian noise) to gradients.
>     *   **State-Dependent:** The collision mechanism is **conditional**. It only activates when neurons exhibit high similarity (i.e., are physically "close" in the phase space).
>     *   **Directional:** The induced updates act as a repulsive force designed to orthogonalize the weight vectors. In the "Soft Collision" variant (Algorithm 2), this process is entirely deterministic. In "Hard Collision," while the scattering angle is stochastic, the energy and momentum conservation laws ensure the updates remain physically constrained.
> 2.  **Constructive Variance:** You are correct that collisions introduce variance, but this variance is **constructive rather than disruptive**. By introducing local variance specifically between highly correlated neurons, KO effectively breaks the symmetry that leads to parameter condensation. As analyzed in Theorem 3.3, this introduced variance is strictly bounded, promoting exploration within the parameter space without violating the conditions required for convergence. Consequently, the **impact** is a proven reduction in weight correlation (Theorem 3.2), which directly translates to tighter generalization bounds (Theorem 3.1) and improved empirical accuracy.
>
> **Q2: I wonder how the 'collision' setup should change across the depth of the neural network, as well as during training. Should more collisions happen earlier during training, or in earlier layers?**
>
> A2: Thank you for this insightful question regarding the spatiotemporal dynamics of the collision mechanism.
>
> 1.  **Spatial Distribution (Depth):**
>     While our main experiments focused on the classifier layer (the final FC layer in ResNets), we conducted an additional controlled experiment to investigate the impact of collision depth. We utilized a 5-layer fully connected network ($5\text{-}50\text{-}50\text{-}50\text{-}50\text{-}1$) on the Multidimensional Synthetic Dataset and applied collisions (coefficient = 0.9) at different depths. The results are summarized below:
>
>     | Collision Location | None | Layer 1 (Early) | Layer 2 (Middle) | Layer 3 (Late) |
>     | :--- | :---: | :---: | :---: | :---: |
>     | **Final Weight Correlation** | 0.7997 | **0.6762** | 0.7888 | 0.7981 |
>
>     **Observation:** Applying collisions at the earliest layer (Layer 1) yields the most significant reduction in weight correlation.
>     **Interpretation:** This supports the hypothesis that parameter condensation has a **cascading effect**. If neurons in early layers collapse into a low-dimensional subspace, this redundancy propagates and amplifies through subsequent layers, limiting the expressivity of the entire network. By intervening early (Layer 1), KO addresses the root cause, fostering diverse feature extraction that benefits all downstream layers.
>
> 2.  **Temporal Distribution (Training Progress):**
>     Regarding the timing, a key advantage of KO is that it is **self-regulating** rather than schedule-based. The collision frequency is determined endogenously by the state of the network:
>     *   **Early Phase:** Condensation pressure is typically highest during the initial training phase (as noted in Zhou et al., 2022a). Consequently, the collision condition (high similarity) is met more frequently, triggering active intervention to break symmetries.
>     *   **Late Phase:** As the network parameters differentiate and stabilize, the collision rate naturally decays.
>
>     This behavior creates an **implicit curriculum**: the optimizer aggressively promotes diversity early on and seamlessly transitions to fine-tuning in later stages, without requiring manual schedule tuning.

---

> > ### Author Response · Authors · 2025-11-25
> >
> > **Q3: The method is argued to increase the entropy of the weights from one step to the next. How does this compare with entropy being used as a regularization term?**
> >
> > A3: Thank you very much for raising this question.
> >
> > 1. **Connection to Regularization**: We interpret "entropy regularization" in this context as methods that enforce diversity or orthogonality among weights (e.g., minimizing weight correlation as proposed in [1]). We view weight correlation as a practical proxy for the entropy measure you mentioned. In our experiments, we directly compared our KO method against this regularization-based approach. As shown in the table below, KO consistently outperforms the regularization baseline on CIFAR-10.
> >
> > 2. **Mechanism Difference** More importantly, [1] addresses correlation only implicitly through a regularization penalty, whereas our KO method tackles neuron condensation directly through the collision mechanism. We summarize the difference as below:
> > * **Regularization ([1])**: This approach modifies the loss landscape by adding a global penalty term. This creates a permanent trade-off between the task loss and the regularization term, effectively applying a static "pressure" on the weights throughout training, even when it might not be necessary.
> > * **KO (Ours)**: Our method modifies the optimization trajectory by an **event-driven** method. We tackle neuron condensation directly via a dynamic collision mechanism that only intervenes when neurons actively approach a condensed state. This allows the optimization to proceed naturally when diversity is high, and applies a "surgical" correction only when condensation is detected
> >
> > By avoiding the modification of the loss function, we avoid creating local minima associated with regularization terms. This allows us to establish clearer theoretical guarantees on convergence and condensation reduction, making KO a more principled and effective solution than indirect regularization.
> >
> > |              | Vanilla      | S.C          | H.C          | regularization[1] |
> > | ------------ | ------------ | ------------ | ------------ | -------------- |
> > | Resnet18     | 95.07% $\pm$ 0.18% | 95.74% $\pm$ 0.05% | 95.58% $\pm$ 0.07%|95.51% $\pm$ 0.23%|
> > | Resnet34     | 95.14% $\pm$ 0.37% |95.76% $\pm$ 0.08%|95.56% $\pm$ 0.14%|95.68% $\pm$ 0.29%|
> > | Resnet50     | 95.37% $\pm$ 0.38% |95.83% $\pm$ 0.16%|95.57% $\pm$ 0.05%|95.42% $\pm$ 0.24%|
> >
> > [1] Jin, G., Yi, X., Zhang, L., Zhang, L., Schewe, S., & Huang, X. (2020). How does weight correlation affect generalisation ability of deep neural networks?. Advances in Neural Information Processing Systems, 33, 21346-21356.
> >
> >
> > **Q4: The improvements are higher in less complex classification problems and also smaller neural networks. How is this justified?**
> >
> > A4: We appreciate this observation. This trend is consistent with the mechanism of our method and can be explained by the relationship between task complexity and model capacity.
> >
> > 1. **Data Complexity v.s. Natural Diversity**: On complex datasets (e.g., ImageNet), the optimization pressure naturally forces the network to utilize a larger portion of its representational capacity to minimize the loss. In this regime, neurons are implicitly encouraged to learn diverse features to handle the complex data distribution, making "natural" condensation less frequent. Conversely, on simpler datasets (e.g., CIFAR), deep networks are often **over-parameterized** relative to the task. This "lazy regime" allows neurons to easily condense or co-adapt without hurting training loss, even though it harms test generalization. Our method (KO) explicitly penalizes this redundancy, providing the largest gains in scenarios where the model would otherwise default to a low-rank, condensed solution.
> >
> > 2. **Network Size and Sensitivity**: Regarding network size, we attribute the larger relative gains in smaller networks to two factors
> > * **Criticality of Capacity**: In smaller networks, representational capacity is a scarce resource. If a subset of neurons condenses, the effective capacity of the model drops significantly, directly impacting performance. By forcing these neurons to remain distinct, our method restores the effective rank of the small network, yielding a substantial boost. In larger networks, there is a "buffer" of redundancy, so a small amount of condensation is less fatal.
> > * **Relative Impact of the Head**: As noted, our method modifies the gradients of the final classification layer. In smaller architectures, this layer often represents a more critical information bottleneck relative to the feature extractor. Therefore, optimizing the geometry of the classification head via collisions has a more pronounced impact on the overall system performance compared to larger, deeper architectures, where the feature extraction capacity is vast.

---

> > > ### Author Response · Authors · 2025-11-25
> > >
> > > **Q5: Are the obtained networks more susceptible to adversarial perturbations?**
> > >
> > > A5: Thank you for raising this critical question regarding robustness. We are happy to report that our method actually **enhances** resilience to adversarial perturbations, rather than increasing susceptibility.
> > >
> > > 1.  **Theoretical Insight:**
> > >     We attribute this improved robustness to diverse feature support. Parameter condensation implies that the network relies on a limited, low-rank set of features. An adversary only needs to perturb inputs along these few directions to fool the model. By enforcing feature diversity via collisions, KO ensures that predictions are supported by a broader, more redundant set of signals, making the decision boundary more robust to single-direction perturbations.
> > >
> > > 2.  **Empirical Verification (FGSM & PGD):**
> > >     We evaluated robustness on CIFAR-10 against Fast Gradient Sign Method (FGSM) [3] and Projected Gradient Descent (PGD) [2] **without any adversarial training**.
> > >     As shown in the table below, KO consistently improves robustness.
> > >     *   **FGSM:** We observe a significant gain (e.g., **+6.17%** on ResNet18 with Hard Collision), confirming resistance to single-step perturbations.
> > >     *   **PGD:** While PGD effectively breaks all non-adversarially trained models (resulting in near-zero accuracy across the board), KO still maintains a slight edge over the baseline.
> > >
> > >     This confirms that mitigating parameter condensation naturally yields a more robust geometric structure in the parameter space.
> > >
> > > |              | FGSM |PGD|
> > > | ------------ | ----- | --- |
> > > | Resnet18     |40.46%|0.08%|
> > > | Resnet18+S.C |43.61%|0.19%|
> > > | Resnet18+H.C |46.63%|0.14%|
> > > | Resnet34     |42.13%|0.04%|
> > > | Resnet34+S.C |44.72%|0.23%|
> > > | Resnet34+H.C |50.08%|0.12%|
> > > | Resnet50     |38.88%|0.00%|
> > > | Resnet50+S.C |39.53%|0.00%|
> > > | Resnet50+H.C |40.21%|0.00%|
> > >
> > > [2] Madry, A., Makelov, A., Schmidt, L., Tsipras, D., & Vladu, A. (2017). Towards deep learning models resistant to adversarial attacks. arXiv preprint arXiv:1706.06083.
> > >
> > > [3] Goodfellow, I. J., Shlens, J., & Szegedy, C. (2014). Explaining and harnessing adversarial examples. arXiv preprint arXiv:1412.6572.

---

### Official Review · Reviewer_kjtd · 2025-11-04

**Soundness:** 2
**Presentation:** 2
**Contribution:** 2
**Rating:** 2
**Confidence:** 3

**Summary:**

This paper proposes a physics-inspired optimizer by considering that weights are particles. The idea is to use particle collision dynamics to modify gradients such that the weights/neuron condensation is reduced. The results show marginal improvements.

**Strengths:**

1. The idea of using particle collision dynamics to reduce weight condensation is interesting and might be impactful.
2. Since the idea only modifies the gradients, it could be used with existing optimizers.
3. The results on classification tasks show marginal but consistent improvements on both validation accuracy and weight condensation.

**Weaknesses:**

1. The method is proposed as a way to reduce neuron condensation; however, the modification is only applied to the gradients and not the updated weights. This is important as weights are initialized randomly, so they are all different, and just using gradient similarity to modify them does not make sense to me. Please clarify this.
2. Weight condensation and neuron condensation are used interchangeably; however, they might mean different things. Please clarify. Also, please clarify how cosine similarity is used as a metric for weight condensation because cosine similarity can only be computed between two vectors.
3. Weight decay might encourage the parameters to become low rank [1], which is contradictory to what is written in line 45.
4. The improvements are marginal, and this casts doubts on how useful the proposed physics-inspired modifications are.

[1] Súkeník, Peter, Christoph Lampert, and Marco Mondelli. "Neural collapse vs. low-rank bias: Is deep neural collapse really optimal?." Advances in Neural Information Processing Systems 37 (2024): 138250-138288.

**Questions:**

1. What is meant by: "Weight correlation is defined as the abstract sum of the network weights’ cosine similarity matrix."? Please provide the equation.

---

> ### Author Response · Authors · 2025-11-25
>
> **Q1: The method is proposed as a way to reduce neuron condensation; however, the modification is only applied to the gradients and not the updated weights.**
>
> A1: Thank you for this important question. The decision to modify gradients instead of weights is deliberate and grounded in both our physical motivation and practical utility:
>
> *   **Physical Consistency:** In our kinetic analogy, gradients represent the **velocity** of parameters (particles). To alter the distribution of particles (weights) and prevent clustering, it is physically more natural to adjust their velocities (via simulated collisions) rather than arbitrarily resetting their positions. This steers the optimization trajectory toward diversity before the step is taken.
> *   **Theoretical Support:** We explicitly prove in **Theorem 3.2** that this gradient-level intervention mathematically necessitates a reduction in the correlation of the updated weights.
> *   **Optimizer Compatibility:** Operating on gradients ensures full compatibility with stateful optimizers like Adam. Direct weight manipulation conflicts with internal states (e.g., momentum buffers), whereas our approach treats the collision as a corrective force that the base optimizer can naturally integrate, ensuring stability.
>
> **Q2: Weights are initialized randomly, so they are all different, and just using gradient similarity to modify them does not make sense to me. Please clarify this.**
>
> A2: Thank you very much for raising this concern. We agree that random initialization provides initial diversity; however, it does not prevent neurons from converging to similar states (condensation) during the training dynamics. We introduce gradient similarity to address this.
>
> 1. **Gradient Similarity isn't the only Evaluation Metric**: To clarify, we do not rely solely on gradient similarity. In soft collision, **both** weight similarity and gradient similarity jointly determine the similarity score between neurons. Here, gradient similarity serves as an additional signal that helps us more reliably detect true weight-level condensation, thereby improving the effectiveness of the collision mechanism.
>
> 2. **Why Gradient Similarity Matters**: Weight distance tells us where neurons are, but gradient similarity tells us where they are going.
> * If two neurons are close in weight space but have different gradients, they will naturally separate in the next step. We should not intervene here.
> * If two neurons are close and have similar gradients, they are "locked" together and will likely condense. This is the specific scenario where our collision mechanism intervenes.
>
> 3. **Functional Redundancy**: Even if weights differ slightly due to random initialization, high gradient similarity indicates that two neurons are reacting to the input data in the exact same way (i.e., they are learning the same feature). By penalizing this gradient alignment, we force the network to learn diverse features rather than redundant ones.
>
> **Q3: Please clarify whether weight condensation or neuron condensation is of concern. Please clarify how cosine similarity is used as a metric for weight condensation.**
>
> A3: Thank you for your insightful question regarding the definition of condensation and the metrics used in our analysis.
>
> **1. Clarification on Neuron Condensation vs. Weight Condensation:**
> We distinguish between the phenomenon of *weight condensation* and the dynamic criteria we use to identify *neuron condensation*.
> *   **Weight Condensation** typically refers to the static scenario where weight vectors cluster into low-dimensional subspaces (high cosine similarity).
> *   **Neuron Condensation:** We argue that weight similarity alone is insufficient to justify intervention. In our framework, we jointly consider **weight similarity and gradient similarity**. The reason is explained above (A2).
>
> **2. Evaluation Metric (Cosine Similarity):**
> To quantitatively evaluate weight condensation, we adopt the weight correlation metric defined in [1] (referenced as Theorem 3.1 in our paper). This metric calculates the average pairwise absolute cosine similarity of the weight vectors. For a layer with $N_l$ neurons, the condensation score $\rho(w_l)$ is defined as:
>
> $$\rho(w_l) =\frac{1}{N_l(N_l-1)} \sum_{i,j=1 \atop i \ne j}^{N_l} \frac{|w_{l i}^T w_{l j}|}{\Vert w_{l i}\Vert_2 \Vert w_{lj}\Vert_2}$$
> The gradient condensation can be calculated similarly.
>
> **References:**
>
> [1] Jin, G., Yi, X., Zhang, L., Zhang, L., Schewe, S., & Huang, X. (2020). How does weight correlation affect generalisation ability of deep neural networks?. *NeurIPS*, 33.

---

> > ### Author Response · Authors · 2025-11-25
> >
> > **Q4: Weight decay might encourage the parameters to become low rank [2], which is contradictory to what is written in line 45.**
> >
> > A4: Thank you for raising this insightful point. We have carefully reviewed Súkeník et al. [2] and conducted an additional ablation study to verify the relationship between Weight Decay (WD) and Condensation. Our findings reveal a nuanced interaction that reconciles our statement in Line 45 with the low-rank bias described in [2].
> >
> > **1. Ablation Study: WD Delays but Does Not Prevent Weight Condensation**
> > To empirically verify the impact of WD, we performed an ablation study on the synthetic dataset (Section 4.1). For simplicity, we only track the weight condensation throughout training:
> >
> > | Weight Decay | Condensation (Init) | Condensation (Mid) | Condensation (End) |
> > | :--- | :--- | :--- | :--- |
> > | **Zero (0)** | 0.3867 | 0.9672 | 0.9999 |
> > | **Standard (1e-3)** | 0.3867 | 0.9676 | 0.9999 |
> > | **Strong (1e-1)** | 0.3867 | **0.9245** | **0.9992** |
> >
> > *   **Transient Mitigation:** As shown in the "Mid" column, strong WD indeed slightly reduces condensation (0.9245 vs. 0.9676). This supports our original observation in Line 45 that regularization can "partially mitigate" the issue by constraining parameter freedom during early-to-mid training.
> > *   **Ultimate Failure:** However, as shown in the "End" column, condensation inevitably reaches extreme levels regardless of WD strength. This aligns perfectly with the theoretical findings in [2] that the optimization landscape inherently favors low-rank solutions (i.e., condensed neurons) upon convergence.
> >
> > **2. Reconciling with [2]: Norm vs. Angle**
> > The contradiction might be resolved by distinguishing between *norms* and *angles*. Weight decay penalizes the magnitude (norm) of weights, which creates a "viscosity" that temporarily hinders neurons from racing towards the same direction (explaining the "Mid" result). However, WD provides no explicit repulsive force to separate their **directions** (angles). Consequently, the implicit low-rank bias of gradient descent, as analyzed in [2], eventually dominates, driving neurons to collapse into identical vectors (explaining the "End" result).
> >
> > In conclusion, while WD offers a minor, temporary resistance to condensation (Line 45), it fails to prevent the ultimate collapse into a low-rank state (as predicted by [2]). This underscores the necessity of our proposed **KO**, which introduces explicit collision dynamics to govern angular diversity, achieving what WD cannot.
> >
> > [2] Súkeník, Peter, Christoph Lampert, and Marco Mondelli. "Neural collapse vs. low-rank bias: Is deep neural collapse really optimal?." Advances in Neural Information Processing Systems 37 (2024): 138250-138288.
> >
> >
> > **Q5: The improvements are marginal, and this casts doubts on how useful the proposed physics-inspired modifications are.**
> >
> > A5: We appreciate the reviewer’s comment regarding the significance of performance gains. While improvement may appear marginal in some classification tasks, we argue that the efficiency implications of such gains are substantial. To demonstrate this scalability and value, we conducted additional pre-training experiments on Large Language Models (LLMs).
> >
> > Specifically, we trained Qwen2-0.5B and Qwen2-1.5B models on the FineWeb-Edu dataset for 50B and 25B tokens, respectively. Our proposed soft collision method (S.C.) was applied to the last transformer layer. We evaluated the models on a diverse set of challenging benchmarks, including Knowledge-intensive tasks (ARC-C, ARC-E) and Commonsense Reasoning (HellaSwag, WinoGrande, OBQA, PIQA).
> >
> > As shown in the table below, our method consistently outperforms the baseline. We achieve an average absolute improvement of >1.4% across all tasks (e.g., +1.6% on Qwen2-1.5B). Crucially, this performance gain is approximately equivalent to a **10% reduction in training steps** to reach baseline accuracy. This benefit significantly outweighs the minor computational overhead (~3.3%) introduced by our method.
> > This provides strong evidence that our approach offers a favorable trade-off between training cost and model performance.
> >
> > | Model \ Acc $\uparrow$ | PIQA      |ARC\_C   | ARC\_E   | HellaSwag | WinoGrande | OBQA | Avg |
> > | -------------------- | --------- |---------| ---------| ---- | ---- | ---- | ---- |
> > | Qwen2-0.5B                 | 0.575     |0.227    | 0.371   |0.273 |0.498 |0.246 | 0.365 |
> > | Qwen2-0.5B-S.C.           | **0.582** |**0.241**|**0.385**|**0.276**|**0.522**| **0.268**    | **0.379** |
> > | Qwen2-1.5B                 | 0.602     |0.236    | 0.439   |0.310 |0.495 |0.256 | 0.390 |
> > | Qwen2-1.5B-S.C.           | **0.615** |**0.258**|**0.450**|**0.313**|**0.535**| **0.266**    | **0.406** |

---

### Official Review · Reviewer_Nk8n · 2025-11-05

**Soundness:** 4
**Presentation:** 2
**Contribution:** 3
**Rating:** 6
**Confidence:** 4

**Summary:**

This paper introduces KO (Kinetics-inspired Optimizer), a neural optimization method motivated by kinetic theory and PDE simulations. The authors provide theoretical analysis and empirical results that together support the effectiveness of the proposed approach.

**Strengths:**

1. The collision-inspired mechanism is conceptually interesting and plausibly bridges ideas from kinetic theory to practical optimization.
2. The paper includes both proofs and experiments, which helps substantiate the method’s validity.

**Weaknesses:**

There are shortcomings in the manuscript’s writing standards and in the completeness of the experiments. See the following questions.

**Questions:**

1. There are severe issues in writing and formatting, for example at lines 164, 177, 215, 246, 248, 295, and 363.

2. In Appendix Table 4, compared with the original network training, the relative time increase for ResNet-34 + H.C. is markedly higher than for ResNet-18 + H.C. and ResNet-50 + H.C., which appears unreasonable; please explain.

3. In Appendix Table 5, why is there no runtime comparison for the CIFAR-100 dataset?

4. How is coll_coef selected in the experiments, and on what basis? Please add ablation studies on hyperparameters, such as coll_coef.

5. In the Related Work section, the literature is discussed only up to 2022 and earlier; please add the latest works. The experiments should include comparisons with methods of the same type; if none are available, please state so.

6. Collision is an interesting idea. If, instead of collisions, one uses short-range repulsion and long-range attraction, how would this compare with collisions? No additional experiments are needed, but please provide a discussion.

7. Please add comparative experiments under initial learning rates of different orders of magnitude, not only the current initial learning rate of 0.1.

---

> ### Author Response · Authors · 2025-11-25
>
> **Q1: Explanation on Appendix Table 4's unreasonable runtime increase and query for runtime comparison for the CIFAR-100 dataset.**
>
> A1: We thank the reviewer for their careful examination of our results and for raising this important point.
>
> 1.  **Correction regarding Appendix Table 4 (ResNet34 Runtime):**
>     Upon re-evaluating our benchmarking environment, we identified that the anomalous runtime increase for ResNet34 in the original submission was caused by resource contention on a shared GPU cluster. We have re-run these experiments in an isolated environment to ensure accuracy. As shown in the revised manuscript (and the table below), the corrected runtime overhead is negligible and consistent with that of other architectures. This confirms that our method scales predictably without the previously observed fluctuations.
>
> 2.  **Runtime Analysis on CIFAR-100:**
>     Per your suggestion, we extended our runtime analysis to the CIFAR-100 dataset. The results demonstrate a clear distinction between the two collision strategies:
>     *   **Hard Collision (H.C.)** suffers from severe runtime degradation (approx. 2–3$\times$ slower), likely due to memory-bound operations that disrupt GPU parallelism.
>     *   **Soft Collision (S.C.)**, in contrast, maintains a runtime almost identical to the Vanilla baseline ($<1\%$ overhead). We attribute this efficiency to Soft Collision's reliance on dense computation operations, which are highly parallelizable and benefit significantly from GPU hardware acceleration.
>
>     The updated statistics (see the Table below) confirm that Soft Collision is a practical, plug-and-play optimization suitable for large-scale models and datasets, offering superior efficiency compared to Hard Collision.
>
>
> | Model | Method | Runtime (s) | Overhead vs. Vanilla |
> | :--- | :--- | :---: | :---: |
> | **ResNet18** | Vanilla | 5,774 | - |
> | | **Ours (S.C.)** | **5,802** | **+0.48%** |
> | | Hard Collision (H.C.) | 16,389 | +183.8% |
> | **ResNet34** | Vanilla | 8,857 | - |
> | | **Ours (S.C.)** | **8,986** | **+1.46%** |
> | | Hard Collision (H.C.) | 18,405 | +107.8% |
> | **ResNet50** | Vanilla | 13,308 | - |
> | | **Ours (S.C.)** | **13,310** | **+0.02%** |
> | | Hard Collision (H.C.) | 26,438 | +98.6% |
>
>
>
>
> **Q2: How is coll_coef selected in the experiments, and on what basis? Please add ablation studies on hyperparameters, such as coll_coef.**
>
> **A2: We thank the reviewer for highlighting the importance of hyperparameter sensitivity.**
>
> 1.  **Selection Basis:**
>     In our experiments, the collision coefficient (`coll_coef`) was selected via a standard grid search on the validation set. Crucially, however, our empirical results indicate that **precise tuning is not strictly necessary**. The introduction of the collision mechanism consistently yields performance gains across a wide range of coefficient values compared to the vanilla baseline.
>
> 2.  **Ablation Study on `coll_coef`:**
>     As requested, we conducted a detailed ablation study on ResNet-50 (CIFAR-10) to evaluate the impact of `coll_coef`. The results, presented in the table below, demonstrate the robustness of our method:
>     *   **Consistent Improvement:** Soft Collision outperforms the baseline (where `coll_coef=0.0`) across the entire spectrum of tested values ($0.1 - 0.9$).
>     *   **Insensitivity:** While the peak performance is observed at `0.5` (+1.54%), even suboptimal values (e.g., `0.9`) provide a significant boost over the baseline (+0.78%).
>
>
>
> | `coll_coef` | **0.0 (Baseline)** | 0.1 | 0.2 | 0.3 | 0.4 | **0.5** | 0.6 | 0.7 | 0.8 | 0.9 |
> | :--- | :---: | :---: | :---: | :---: | :---: | :---: | :---: | :---: | :---: | :---: |
> | **Accuracy** | 77.38% | 78.23% | 78.80% | 78.30% | 77.98% | **78.92%** | 77.52% | 77.87% | 78.01% | 78.16% |
> | **Gain** | - | +0.85 | +1.42 | +0.92 | +0.60 | **+1.54** | +0.14 | +0.49 | +0.63 | +0.78 |
>
> ---

---

> > ### Author Response · Authors · 2025-11-25
> >
> > **Q3: Please add the latest works. The experiments should include comparisons with methods of the same type.**
> >
> > A3: We thank the reviewer for this constructive suggestion. We have incorporated the suggested references and expanded our comparison to include relevant baselines.
> >
> > 1.  **Distinctions from Recent Literature:**
> >     To the best of our knowledge, direct optimization based on *intra-layer neuron similarity* remains under-explored. While recent works touch upon similar concepts, they differ fundamentally in their focus:
> >     *   **Neural Collapse (Feature Space) [2, 3]:** These works analyze the "Neural Collapse" phenomenon, where feature embeddings of the same class converge. This line of research focuses on the geometry of **activations and outputs**. In contrast, our work targets **weight-space similarity**, which is independent of the input data and focuses on the structural diversity of neurons themselves.
> >     *   **Cross-Model Analysis (Weight Space) [4]:** While [4] examines weight similarity, it does so across *different* networks to analyze representation alignment. It does not intervene during training. Our method, conversely, is an active optimization strategy applied within a single network during training.
> >
> > 2.  **Comparison with the Closest Baseline:**
> >     The most relevant baseline is **[1]**, which introduces a weight decorrelation term as a regularization penalty (Soft Constraint). We compared this method against our proposed KO optimizer (specifically Soft Collision). As shown in the table below, **KO consistently outperforms the regularization baseline [1] on CIFAR-10.**
> >
> >     **Why our method is superior:**
> >     The advantage of KO stems from its optimization mechanism.
> >     *   **Method [1] (Regularization):** Treats correlation as a penalty term added to the loss. This creates a trade-off where the optimizer must balance reducing the primary loss against the regularization term, often leading to suboptimal convergence or hyperparameter sensitivity.
> >     *   **Ours (Collision):** Addresses weight condensation **directly via the update step**. By explicitly modifying the gradient direction to avoid collisions, we provide a theoretical guarantee that weight correlation is reduced relative to the original optimizer, *without* compromising the convergence speed of the main task.
> >
> >     Empirically, this results in higher accuracy and lower variance (better stability) compared to [1].
> >
> > | Model | Vanilla | **Ours (Soft Collision)** | Ours (Hard Collision) | Regularization [1] |
> > | :--- | :--- | :--- | :--- | :--- |
> > | **ResNet18** | 95.07% ($\pm$ 0.18) | **95.74% ($\pm$ 0.05)** | 95.58% ($\pm$ 0.07) | 95.51% ($\pm$ 0.23) |
> > | **ResNet34** | 95.14% ($\pm$ 0.37) | **95.76% ($\pm$ 0.08)** | 95.56% ($\pm$ 0.14) | 95.68% ($\pm$ 0.29) |
> > | **ResNet50** | 95.37% ($\pm$ 0.38) | **95.83% ($\pm$ 0.16)** | 95.57% ($\pm$ 0.05) | 95.42% ($\pm$ 0.24) |
> >
> >
> >
> > [1] Jin, G., Yi, X., Zhang, L., Zhang, L., Schewe, S., & Huang, X. (2020). How does weight correlation affect generalisation ability of deep neural networks?. Advances in Neural Information Processing Systems, 33, 21346-21356.
> >
> > [2] Andriopoulos, G., Dong, Z., Guo, L., Zhao, Z., & Ross, K. (2024). The prevalence of neural collapse in neural multivariate regression. Advances in Neural Information Processing Systems, 37, 126417-126451.
> >
> > [3] Kothapalli, V. (2022). Neural collapse: A review on modelling principles and generalization. arXiv preprint arXiv:2206.04041.
> >
> > [4] Min, Z., & Wang, X. (2025). DOCS: Quantifying weight similarity for deeper insights into large language models. arXiv preprint arXiv:2501.16650.

---

> > > ### Author Response · Authors · 2025-11-25
> > >
> > > **Q4: If, instead of collisions, one uses short-range repulsion and long-range attraction, how would this compare with collisions?**
> > >
> > > A4: We thank the reviewer for this insightful analogy connecting our method to physics-inspired dynamics.
> > >
> > > 1.  **Short-range Repulsion $\equiv$ Soft Collision:**
> > >     We agree with the reviewer's observation. Our **Soft Collision** mechanism is functionally equivalent to **short-range repulsion**. It operates on the principle that when neurons become excessively similar (i.e., "too close" in weight space), a repulsive force is needed to push them apart.
> > >     *   **Soft Collision:** Acts as a smooth repulsive potential (e.g., electrostatic repulsion).
> > >     *   **Hard Collision:** Can be viewed as the limit case of repulsion (infinite potential barrier at contact).
> > >
> > >     Both strategies effectively prevent *Neuron Condensation* by ensuring distinct representations.
> > >
> > > 2.  **Long-range Attraction $\approx$ Standard Gradient Descent:**
> > >     Regarding "long-range attraction," we argue that **standard task-driven training inherently provides this force**.
> > >     *   The loss function pulls neurons toward useful features in the optimization landscape.
> > >     *   When multiple neurons are attracted to the same optimal feature, they tend to cluster together (the root cause of condensation).
> > >     *   Therefore, adding an *explicit* inter-neuron attraction term is unnecessary and might even exacerbate the condensation problem we aim to solve.
> > >
> > > 3.  **Conclusion: KO integrates both forces.**
> > >     Our Kinetic Optimizer (KO) can be viewed as a unified implementation of the dynamics suggested by the reviewer:
> > >     *   **Attraction:** Provided by the standard gradient (pulling weights to valid solutions).
> > >     *   **Repulsion:** Provided by the Collision term (maintaining diversity).
> > >
> > >     Thus, KO already achieves the balance between "short-range repulsion and global attraction," while providing rigorous theoretical guarantees and high computational efficiency.
> > >
> > > **Q5: Please add comparative experiments under initial learning rates of different orders of magnitude, not only the current initial learning rate of 0.1.**
> > >
> > > A5:
> > > Thank you for the constructive comment. Following your suggestion, we conducted additional experiments using a wider range of initial learning rates ($0.05, 0.1, 0.5$) on CIFAR-10 to verify the method's generalization ability.
> > >
> > > As shown in the following table, our proposed Soft Collision (S.C.) and Hard Collision (H.C.) mechanisms consistently surpass the Vanilla baseline in all settings. Notably, even at a suboptimal learning rate of 0.5, our approach achieves a performance gain of approximately $0.4\%$ compared to the baseline. Furthermore, the improvements remain significant at a smaller learning rate of 0.05 (e.g., ResNet50+S.C. reaches $95.87\%$).
> > >
> > > These results confirm that our method is robust to initial learning rate changes and does not rely on extensive tuning to achieve performance improvements.
> > >
> > >
> > >
> > > | Model Setting | lr=0.1 | lr=0.5 | lr=0.05 |
> > > | :--- | :---: | :---: | :---: |
> > > | **ResNet18 (Baseline)** | 95.07% $\pm$ 0.18% | 92.71% $\pm$ 0.24% | 95.32% $\pm$ 0.12% |
> > > | + S.C. | **95.74% $\pm$ 0.05%** | **93.18% $\pm$ 0.08%** | **95.47% $\pm$ 0.03%** |
> > > | + H.C. | 95.58% $\pm$ 0.07% | 92.77% $\pm$ 0.12% | 95.33% $\pm$ 0.09% |
> > > | **ResNet34 (Baseline)** | 95.14% $\pm$ 0.37% | 93.29% $\pm$ 0.42% | 95.40% $\pm$ 0.27% |
> > > | + S.C. | **95.76% $\pm$ 0.08%** | **93.52% $\pm$ 0.20%** | **95.69% $\pm$ 0.06%** |
> > > | + H.C. | 95.56% $\pm$ 0.14% | 93.44% $\pm$ 0.29% | 95.47% $\pm$ 0.11% |
> > > | **ResNet50 (Baseline)** | 95.37% $\pm$ 0.38% | 93.26% $\pm$ 0.37% | 95.42% $\pm$ 0.07% |
> > > | + S.C. | **95.83% $\pm$ 0.16%** | **93.61% $\pm$ 0.18%** | **95.87% $\pm$ 0.06%** |
> > > | + H.C. | 95.57% $\pm$ 0.05% | 93.44% $\pm$ 0.09% | 95.41% $\pm$ 0.04% |
> > >
> > > **Q6: Issues in writing and formatting**
> > >
> > > A6: Thank you very much for such a thorough review. In the updated draft, we have corrected all the mentioned writing and formatting problems. We apologize for any misunderstanding caused by those writing mistakes.

---

### Meta-Review · Area_Chair_LLuk · 2026-01-07

**Summary:**

The paper proposes KO, a physics-inspired optimizer that augments standard gradient-based optimizers (e.g., SGD, Adam) with a collision mechanism derived from kinetic theory and numerical schemes for the Boltzmann transport equation. Network parameters are treated as particles, and neuron/weight condensation (the collapse of weights into low-dimensional subspaces) is modeled as excessive similarity between particles. KO modifies gradients (rather than weights directly) to simulate repulsive collisions when neurons become too similar, aiming to preserve parameter diversity while retaining convergence guarantees.

The paper claims three main contributions: (I) A novel optimizer design grounded in kinetic theory with both “soft” (deterministic, matrix-based) and “hard” (stochastic, N-body–like) collision variants. (II) A theoretical analysis linking reduced weight correlation to improved generalization and providing convergence guarantees despite added variance. (III) Empirical results on image and text classification (CIFAR-10/100, ImageNet, IMDB, Snips), with additional late-stage experiments on LLM pretraining, showing modest but consistent accuracy improvements at relatively small computational overhead.

**Reviewer Concerns:**

Initial reviewer opinions were mixed. The raised concerns were as follows:

- Presentation (raised by Nk8n, kjtd, Xj7q): Several reviewers found the paper difficult to read, with vague terminology (parameter diversity, condensation), inconsistent use of weight vs. neuron condensation, missing equations, and unclear mapping between physics concepts and optimization mechanics. Reviewer Xj7q explicitly doubted that they could implement the method from the description. The authors substantially expanded definitions in the rebuttal stage, added formal equations for condensation metrics, and included an explicit analogy table mapping particles to neurons, velocity to gradients, etc. This massively improves clarity (provided that it should make its way into the manuscript itself), and the authors should be commended for it.

- Gradient-level “collisions” (raised by kjtd): The question of why one should operate on gradients rather than weights was raised, especially given that random initialization already provides diversity. Reviewers broadly questioned whether gradient similarity is a sensible proxy and whether the physical analogy is more than metaphorical. The authors argue that gradients correspond to velocities in the kinetic analogy, modifying gradients preserves compatibility with stateful optimizers, and joint consideration of weight and gradient similarity allows intervention only when neurons are both close and moving together. They also provide theoretical arguments that gradient-level intervention implies reduced weight correlation after updates. Provided that one accepts the kinetic analogy, this response is adequate, although a skeptical viewpoint will still view the analogy as heuristic.

- Novelty (raised by uF9q, Nk8n, Xj7q): Several reviewers questioned whether KO is meaningfully different from existing regularization-based approaches (e.g. weight decorrelation, entropy regularization, weight decay) and whether the paper overstates novelty by framing regularization as merely post-hoc. The authors added explicit comparisons to a weight decorrelation baseline and additional ablations with and without weight decay. They argue that KO differs fundamentally by modifying the optimization trajectory rather than the loss landscape and by intervening only when condensation is detected. The empirical evidence here is appreciated, although the claims that the mechanism is fundamentally different from a form of adaptive regularization remains questionable to me.

- Significance of empirical gains (raised by kjtd, Xj7q): The improvements on standard benchmarks are quite small, about 0.3-0.6%, raising doubts about the practical impact relative to added complexity. The authors respond by arguing that small gains are meaningful in mature benchmarks (I am highly skeptical of this claim), and that adding LLM pretraining experiments show 1.4% average gains on downstream evaluations, framed as equivalent to substantial extra training compute. The LLM results strengthen the case for scalability, but are limited in scope and only arrive in the discussion period. More experiments of this nature, along with a proper assessment of their quality, would be required to address this concern.

- Computational cost (raised by Nk8n, Xj7q): It is unclear to the reviewers whether the computational scaling is prohibitive, especially for hard collisions, as there are inconsistent runtime results in the original submission. The authors corrected the runtime anomalies, added CIFAR-100 timing, and clearly distinguished soft (GPU-friendly, matrix-multiplication-based) from hard (memory-bound, cubic scaling) collisions. They effectively concede that hard collisions are impractical at scale. This concern is well addressed and clearly positions soft collision as the intended practical method.

- Contradictions with prior work (raised by kjtd, Xj7q): Some reviewers raised potential contradictions with the literature which suggests low-rank bias or condensation can aid generalization, and that the impact of added variance on convergence is unclear. The authors carefully distinguish weight condensation from feature collapse and argue that their method preserves capacity rather than harming it. They provide bounded-variance arguments and empirical robustness checks. It is unclear to me whether these concerns have been addressed.

**Reviewer Scores:**

The author responses were generally of a high quality and addressed several major concerns. However, the lack of clarity in the initial submission typically hinders further discussion in my experience, and there remain several issues that I am unsure would have been considered addressed by the reviewers. The significance of the empirical gains seems like a major flaw in my view, and the authors should try to design more experiments to address this.

---

### Decision · Program_Chairs · 2026-01-26

Reject